# Ciliary Hedgehog signaling patterns the digestive system to generate mechanical forces driving elongation

Ying Yang[1], Pekka Paivinen [2], Chang Xie[1], Alexis Leigh Krup [1], Tomi P. Makela [2], Keith E. Mostov [1,3] & Jeremy F. Reiter [1,4✉]

How tubular organs elongate is poorly understood. We found that attenuated ciliary Hedgehog signaling in the gut wall impaired patterning of the circumferential smooth muscle and inhibited proliferation and elongation of developing intestine and esophagus. Similarly, ablation of gut-wall smooth muscle cells reduced lengthening. Disruption of ciliary Hedgehog signaling or removal of smooth muscle reduced residual stress within the gut wall and decreased activity of the mechanotransductive effector YAP. Removing YAP in the mesenchyme also reduced proliferation and elongation, but without affecting smooth muscle formation, suggesting that YAP interprets the smooth muscle-generated force to promote longitudinal growth. Additionally, we developed an intestinal culture system that recapitulates the requirements for cilia and mechanical forces in elongation. Pharmacologically activating YAP in this system restored elongation of cilia-deficient intestines. Thus, our results reveal that ciliary Hedgehog signaling patterns the circumferential smooth muscle to generate radial mechanical forces that activate YAP and elongate the gut.

[1] Department of Biochemistry and Biophysics, Cardiovascular Research Institute, University of California San Francisco, San Francisco, CA, USA. [2] iCAN Digital Precision Cancer Medicine Flagship, Research Programs Unit, Faculty of Medicine and HiLIFE-Helsinki Institute of Life Science, University of Helsinki, Helsinki, Finland. [3] Department of Anatomy, University of California San Francisco, San Francisco, CA, USA. [4] Chan Zuckerberg Biohub, San Francisco, CA, USA. ✉email: jeremy.reiter@ucsf.edu

Many of our organs, including the alimentary tract, are epithelial tubes surrounded by mesenchyme. In humans, the endodermal tube post-gastrulation is approximately 1 mm long, and, at birth, the alimentary tract is over 3 m long, representing a > 3,000-fold increase in length[1]. In contrast, the fetus itself elongates approximately 500-fold in the same time. In the postnatal intestine, applied distractive forces can lead to lengthening[2]. The sources of mechanical force in development and whether mechanical force contributes to developmental elongation have been unclear.

Mechanical forces during development can be generated by smooth muscle. During late gestation, smooth muscle generates biophysical cues that instruct the formation of the second longitudinal smooth muscle layer in the intestine[3]. Smooth muscle is also implicated in airway branching[4]. In the mouse gut, smooth muscle arises from mesoderm-derived stromal cells beginning at embryonic day (E) 11. During E12-13, these smooth muscle cells organize into a distinct circumferential layer[5]. The formation of this smooth muscle layer requires Hedgehog (HH) signaling[3,6–9]. Interestingly, Hedgehog (HH) ligands also participate in intestinal lengthening[6,10]. How HH signaling drives both smooth muscle differentiation and intestinal elongation and whether these two events are coupled have not been understood.

Mechanical stimuli can activate the Hippo pathway and lead to organ growth[11,12]. The primary transcriptional effectors of the Hippo pathway are Yes-associated protein (YAP, also called YAP1) and its related co-transcriptional factor TAZ (also called WWTR1)[13,14]. Different mechanical influences cause YAP and TAZ to localize to either the nucleus or the cytoplasm[15,16]. Nuclear YAP and TAZ promote cell proliferation and inhibit apoptosis[17,18]. Although YAP restricts the HH-dependent differentiation of smooth muscle[7], it has been unclear whether, conversely, HH signaling regulates YAP activity.

Vertebrate HH signaling is transduced by a specialized signaling organelle called the primary cilium. Most mammalian cells possess a single, non-motile primary cilium protruding from the cell surface. The primary cilium is comprised of a microtubule-based axoneme built atop a mother centriole and sheathed by a ciliary membrane enriched in specific signaling proteins[19]. HH signals trigger the entry of a seven-pass transmembrane protein, Smoothened (SMO), into the cilium where it activates the downstream transcriptional effectors of the HH pathway[20]. Although cilia transduce HH signals to pattern the limb and neural tube[21–24], whether cilia function in gut development has been unexplored.

In this work, we examined the development of the gut in mouse mutants with defective ciliary signaling and found that cilia are essential to regulate the formation of the circumferential smooth muscle to drive the elongation of developing intestine and esophagus.

## Results

### Cilia are critical for the elongation of tubular organs. Human mutations in Ciliogenesis associated kinase 1 (CILK1, also called Intestinal cell kinase or ICK) cause lethal endocrine-cerebro-osteodysplasia and short rib polydactyly syndromes[25–28]. We and others found that CILK1 is a ciliary kinase important for ciliary morphology and HH signaling[29–32]. To study the effects on intestinal development by impaired cilia, we generated mouse germline and conditional null alleles of Cilk1 (Supplementary Fig. 1A). Consistent with previous reports[30,31], Cilk1−/− embryos survived until birth and displayed edema, shortened ribs and polydactyly (Supplementary Fig. 1B). These phenotypes are consistent with developmental roles for CILK1 in regulating ciliogenesis and ciliary HH signaling. In addition to previously

described phenotypes, we observed that Cilk1−/− embryos exhibited dramatically shortened small intestines (Fig. 1A–B). An approximately 50% reduction in intestinal length was completely penetrant in Cilk1−/− from E14.5 (Fig. 1A–B). Other growth parameters, such as intestinal diameter or body length, were unchanged in Cilk1−/− embryos (Supplementary Fig. 1C–D), indicating that the intestinal shortening phenotype is not reflective of global growth defects.

Quantifying mitotic intestinal cells using immunofluorescence for phosphorylated histone H3 (pHH3) revealed that overall proliferation was reduced in E13.5 Cilk1−/− intestines (Supplementary Fig. 1E, F). Co-staining for α-tubulin-containing mitotic spindles distinguished radially and longitudinally dividing cells. Interestingly, the number of radially dividing cells was not affected by loss of CILK1 (Supplementary Fig. 1E, F). Thus, the reduced elongation of Cilk1−/− intestine is associated with decreased proliferation.

To investigate another part of the gut, we examined the developing esophagus. Like the small intestines, Cilk1−/− esophagi were shorter than those of control at E16.5 (Fig. 1C, D). To test whether, as in the intestine, the reduced elongation of Cilk1−/− esophagi was associated with decreased proliferation, we quantified 5-ethynyl-2′-deoxyuridine (EdU) incorporation into Cilk1−/− and littermate control Cilk+/- esophagi at E14.5, a timepoint prior to the emergence of differences in esophageal length. The proportion of EdU-labeled cells was reduced in Cilk1−/− esophagi (Supplementary Fig. 1G). Thus, CILK1 promotes elongation in multiple parts of the developing gut through proliferation.

To begin to investigate how CILK1 regulates proliferation and elongation, we examined the intestinal cilia in Cilk1 mutants to assess whether CILK1 regulates the composition of cilia in the developing intestine. A ciliary protein, Intraflagellar Transport 88 (IFT88), was distributed along the length of control cilia. In contrast, IFT88 abnormally accumulated in the ciliary tips in both Cilk1−/− intestine epithelial (Fig. 1E, gray box) and mesenchymal cells (Fig. 1E, white box). This altered distribution of IFT88 indicates that CILK1 controls ciliary composition in the developing gut. As IFT88 is critical for intraciliary trafficking and ciliary signal transduction, these results further raise the possibility that the requirement for of CILK1 in tubular elongation could be dependent on cilia.

To test whether the role of CILK1 in intestinal development reflects its ciliary function, we examined intestines of other mutants affecting ciliary function, including mouse embryos lacking B9 domain-containing 1 (B9d1)[33], or Inositol polyphosphate 5-phosphatase (Inpp5e)[34]. In addition, we generated a loss-of-function allele of Tectonic 3 (Tctn3) (Supplementary Fig. 2A), encoding a component of the ciliary transition zone we previously identified[35]. Consistent with a previous report[36], the Tctn3−/− embryos displayed microphthalmia and polydactyly at E13.5 (Supplementary Fig. 2B).

We investigated the B9d1, Inpp5e, and Tctn3 mouse mutants because the homozygous null mutants survive to gestational ages at which intestinal length can be ascertained and because they encode proteins that participate in ciliary signaling through distinct mechanisms. INPP5E generates a cilium-enriched phosphoinositide important for ciliary signaling[37–39]. B9D1 and TCTN3 are components of the ciliary transition zone controlling ciliary composition[35]. Consistent with a function of TCTN3 at the transition zone, immunofluorescence staining for ARL13B, a transition zone-dependent ciliary component, revealed that E13.5 Tctn3−/− embryos displayed reduced numbers of ARL13B-positive cilia (Supplementary Fig. 2C).

Like Cilk1 null mutants, B9d1, Inpp5e, and Tctn3 mutants all exhibited shortened small intestines (Supplementary Fig. 2D–F).

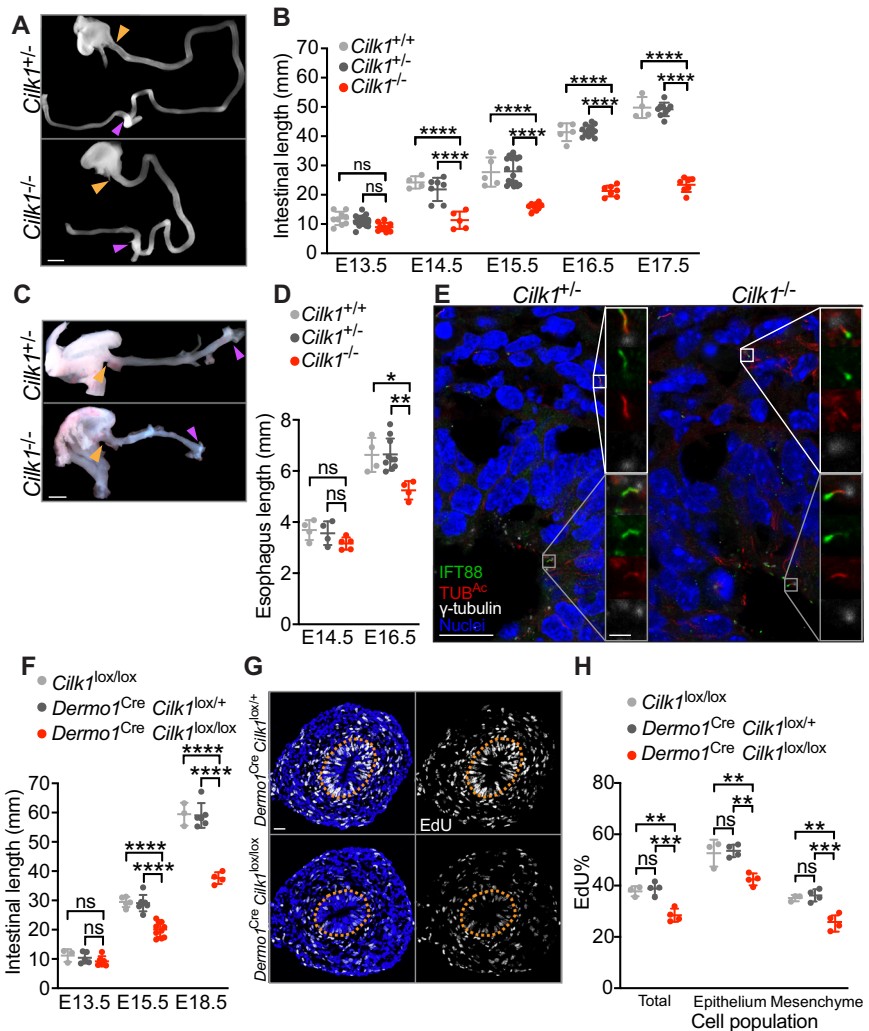

**Fig. 1 Mesenchymal cilia are essential for the cell proliferation and elongation of tubular organs. A** E14.5 intestines after separation from the mesentery of *Cilk1+/−* and *Cilk1−/−* embryos. The length of the small intestine was measured from the caudal stomach (orange arrowheads) to the rostral cecum (purple arrowheads). Scale bar, 1 mm. **B** Lengths of the small intestines of *Cilk1+/+* (n = 9, 4, 5, 5, 4), *Cilk1+/−* (n = 18, 7, 18, 12, 9), and *Cilk1−/−* (n = 10, 5, 9, 7, 7) embryos at E13.5, 14.5, 15.5, 16.5, and 17.5. **C** E16.5 esophagi of *Cilk1+/−* and *Cilk1−/−* embryos. The length of the esophagi was measured from the stomach (orange arrowheads) to the pharynx (purple arrowheads). Scale bar, 1 mm. **D** Lengths of the esophagi of *Cilk1+/+* (n = 4 and 4), *Cilk1+/−* (n = 4 and 8), and *Cilk1−/−* (n = 5 and 4) embryos at E14.5 and E16.5 (adjusted P values are 0.0021 for *Cilk1+/+* vs *Cilk1−/−* and 0.0003 for *Cilk1+/−* vs *Cilk1−/−*). **E** Immunofluorescence staining of E13.5 intestinal cross sections from *Cilk1+/−* and *Cilk1−/−* embryos for IFT88 (green), ciliary axonemes (TUB$^{Ac}$, red), basal bodies (γ-tubulin, white), and nuclei (Hoechst, blue). Right, higher magnifications of boxed cilia. White boxes highlight mesenchymal cilia and gray boxes highlight epithelial cilia. Scale bar of larger images, 5 μm. Scale bar of enlarged images, 2 μm. **F** Lengths of E13.5, E15.5, and E18.5 control (*Cilk1^lox/lox* n = 3, 5, 3 and *Dermo1^Cre Cilk1^lox/+* n = 5, 7, 6) and *Dermo1^Cre Cilk1^lox/lox* (n = 6, 10, 4) small intestines. **G** Immunofluorescence staining for EdU (white) and nuclei (Hoechst, blue) in E13.5 *Dermo1^Cre Cilk1^lox/+* and *Dermo1^Cre Cilk1^lox/lox* intestines. Scale bar, 25 μm. Orange dotted lines outline the epithelium. **H** The percentage of EdU-positive cells in the whole intestine, the epithelial cells and the mesenchymal cells in *Cilk1^lox/lox* (n = 3), *Dermo1^Cre Cilk1^lox/+* (n = 4) and *Dermo1^Cre Cilk1^lox/lox* (n = 4) embryos at E13.5. For each intestine, we averaged the percentages from three sections. **B, D, F**, and **H** Each point in the scatter plots represents the value from an individual embryonic intestine or esophagus. Horizontal bars indicate means ± SD. ns, p > 0.05; *p < 0.05; **p < 0.01; ***p < 0.001 and ****p < 0.0001 indicate adjusted P values between indicated samples by two-way ANOVA Tukey's multiple comparisons test. Source data are provided in the Source Data file.

Thus, disrupting multiple aspects of ciliary function all resulted in shortened intestines, indicating that cilia promote elongation of the gut.

**Mesenchymal cilia drive HH signaling and proliferation.** Mesenchymal cells, which are the main HH responsive cells in the developing gut[40], possessed cilia throughout development (Supplementary Fig. 3A, white boxes). In contrast, epithelial cells, which produce the HH ligands[41], possessed cilia at E13.5 (Fig. 1E) but not at E18.5 (Supplementary Fig. 3A, gray boxes), consistent with a previous study[42]. As cilia were present in both the

endoderm-derived epithelium and the mesoderm-derived mesenchyme surrounding the epithelium during early intestinal development, roles for cilia in intestinal elongation could reflect functions in the epithelium, the mesenchyme or both compartments. To distinguish between these possibilities, we removed CILK1 from either the epithelium or the mesenchyme and assessed intestinal length.

To remove CILK1 specifically from the epithelium, we generated *Shh^Cre Cilk1^lox/lox* embryos, in which *Shh^Cre* [43] deleted *Cilk1* specifically in the endodermal epithelium (Supplementary Fig. 3B). Consistent with paracrine HH signaling in the

intestine[40], the expression of HH target genes *Gli1* and *Ptch1* was enriched in the stromal compartment (Supplementary Fig. 3B). *Gli1* and *Ptch1* expression levels were unchanged in *Shh*^Cre *Cilk1*^lox/lox embryos, and abrogating *Cilk1* expression in the intestinal epithelium did not alter the length of *Shh*^Cre *Cilk1*^lox/lox intestines at E18.5 (Supplementary Fig. 3B, C), suggesting that CILK1 in the epithelium is dispensable for intestinal elongation.

To remove CILK1 specifically from the mesenchyme, we generated *Dermo1*^Cre *Cilk1*^lox/lox embryos. *Dermo1*^Cre [44] was active in mesoderm-derived intestinal mesodermal cells, as revealed by Cre-dependent tdTomato expression (Supplementary Fig. 4A). In *Dermo1*^Cre *Cilk1*^lox/lox intestine, the expression of *Cilk1* was reduced as were the HH target genes *Gli1* and *Ptch1* (Supplementary Fig. 4B), indicating that CILK1 functions in the mesenchyme to transduce HH signals. These results demonstrate that mesenchymal cilia interpret HH signals in the developing gut. Unlike *Shh*^Cre *Cilk1*^lox/lox embryos (Supplementary Fig. 3C), but like *Cilk1*^−/− embryos (Fig. 1A, B), *Dermo1*^Cre *Cilk1*^lox/lox embryos exhibited shortened intestines (Fig. 1F). Thus, intestinal lengthening during development depends upon CILK1 in the mesoderm-derived mesenchyme.

As in the intestine, depletion of CILK1 in the mesoderm-derived cells of *Dermo1*^Cre *Cilk1*^lox/lox embryos shortened esophagi at E15.5 (Supplementary Fig. 4C). Thus, the lengthening of esophagus during development also depends upon ciliary signaling in the mesenchyme.

As CILK1 is required in the mesenchyme for both HH signal transduction and elongation, we hypothesized that ciliary HH signaling in the mesenchyme is essential for intestinal elongation. To test this hypothesis, we removed IFT88, essential for cilia assembly and maintenance[45], in the mesenchyme (by generating *Dermo1*^Cre *Ift88*^lox/- embryos). *Dermo1*^Cre *Ift88*^lox/- embryos, like *Dermo1*^Cre *Cilk1*^lox/lox embryos, exhibited shortened intestines at E14.5 (Supplementary Fig. 4D). Previous work demonstrated that depletion of epithelial HH ligands shortens intestinal length[6]. To test whether HH signals transduced by mesenchymal cilia promote elongation, we removed Smoothened (SMO), an essential component of the HH signal transduction pathway, from the mesenchyme. After generating *Dermo1*^Cre *Smo*^lox/lox embryos and assessing their intestinal length at E14.5, we found that these intestines were shortened by approximately 55% (Supplementary Fig. 4E). Together, these results indicate that mesenchymal cilia transduce HH signals to drive elongation.

How might mesenchymal ciliary signaling promote elongation? As in *Cilk1*^−/− esophagi (Supplementary Fig. 1G), the proportion of EdU-labeled cells was also reduced in *Dermo1*^Cre *Cilk1*^lox/lox intestines (Fig. 1G, H). Interestingly, the decreased EdU labeling in *Dermo1*^Cre *Cilk1*^lox/lox intestines was not restricted to the mesenchyme where CILK1 was deleted, but was also observed in the epithelium (Fig. 1G, H). These results demonstrate that mesenchymal CILK1 is essential for proliferation of both the mesenchyme and its associated epithelium, suggesting that the intestinal mesenchyme has important roles in promoting both proliferation and elongation in both cell compartments.

In addition to proliferation, programmed cell death also plays a role in regulating organ size[46–50]. To assess whether increased apoptosis could account for decreased intestinal elongation, we quantified the prevalence of apoptotic cells marked by cleaved Caspase-3. We detected no increased in apoptosis in *Dermo1*^Cre *Cilk1*^lox/lox guts (Supplementary Fig. 5A, B). Thus, CILK1 is essential for proliferative elongation of the gut tube.

**Ciliary signaling patterns the circumferential smooth muscle.** As HH signaling has been shown to regulate the patterning of smooth muscle cells[3,6–9], we explored whether cilia participate in intestinal smooth muscle development. We stained for α-smooth muscle actin (αSMA), a marker of smooth muscle, at E13.5, prior to altered gut elongation in cilia mutants. We found that at E13.5 smooth muscle organization was disrupted in *Cilk1*^−/− gut, as measured by increased variance of tangential alignment and increased radial distribution (Fig. 2A, B). This patterning defect was not due to the presence of more smooth muscle cells (Fig. 2C), but altered distribution in the gut wall (Fig. 2B). Therefore, CILK1 is not essential for smooth muscle differentiation or survival, but rather for its patterning and spatial organization.

To test whether the role of CILK1 in smooth muscle development is shared by other cilia-related proteins, we examined smooth muscle in *Tctn3*^−/− gut. As with CILK1, TCTN3 was required for organization of the circumferential smooth muscle (Supplementary Fig. 6A, B). Together, these results indicate that cilia are required for the formation of the circumferential smooth muscle in the developing gut.

To assess whether mesenchymal cilia are specifically required for smooth muscle development, as they are for elongation, we examined smooth muscle of E13.5 in control and *Dermo1*^Cre *Cilk1*^lox/lox guts (Supplementary Fig. 6C, D). Similar to *Cilk1*^−/− smooth muscle, the smooth muscle of *Dermo1*^Cre *Cilk1*^lox/lox guts was disorganized with increased radial distribution and disrupted circumferential alignment, indicating that cilia function specifically in the mesenchyme to pattern smooth muscle.

To examine whether mesenchymal ciliary signaling also patterns esophageal smooth muscle, we stained for αSMA in esophagi at E14.5, a timepoint before length is altered in *Cilk1*^−/− embryos. As reported before[3,51], at E14.5, αSMA was predominantly expressed by two layers of esophageal smooth muscle (Fig. 2D upper panel): lamina muscularis interna (purple arrowhead) and lamina muscularis externa (orange arrowhead). Both layers of lamina muscularis surround the esophageal epithelium (Fig. 2D upper panel). In *Cilk1*^−/− esophagi, the spatial separation of the two smooth muscle layers was disrupted (Fig. 2D lower panel and Fig. 2E). This phenotype seemed not due to delayed development, because the two layers in E15.5 *Dermo1*^Cre *Cilk1*^lox/lox esophagi were still not separated as those at E14.5 (Supplementary Fig. 6E, F). In addition, some smooth muscle fibers were misoriented in *Dermo1*^Cre *Cilk1*^lox/lox esophagus (Supplementary Fig. 6E lower panel). Thus, as in the intestine, CILK1 also regulates the spatial organization of smooth muscle in the developing esophagus.

To test whether disrupting SMO also affects smooth muscle development, we analyzed smooth muscle in *Dermo1*^Cre *Smo*^lox/lox and control embryos at E13.5. In *Dermo1*^Cre *Smo*^lox/lox embryos, the circumferential smooth muscle layer was thinner (Supplementary Fig. 7A, B) and the number of αSMA-expressing cells was reduced (Supplementary Fig. 7C). Thus, both cilia and HH signaling are essential to pattern the intestinal circumferential smooth muscle.

**Smooth muscle-generated residual stress drives elongation.** Do disruptions in mesenchymal cilia or HH signaling independently impair smooth muscle development and decrease elongation, or do the defects in smooth muscle cause decreased elongation? To distinguish between these two possibilities, we tested whether smooth muscle itself is critical for elongation. Specifically, we partially ablated smooth muscle by expressing attenuated diphtheria toxin (DTA176) in smooth muscle under the control of the *Myosin heavy polypeptide 11* (*Myh11*) regulatory elements (*Myh11*^Cre-EGFP *R26*^DTA176/+ embryos). To validate decreased smooth muscle, we co-stained the E15.5 intestine for αSMA and GFP. In the *Myh11*^Cre-EGFP intestine, most αSMA-expressing cells also expressed

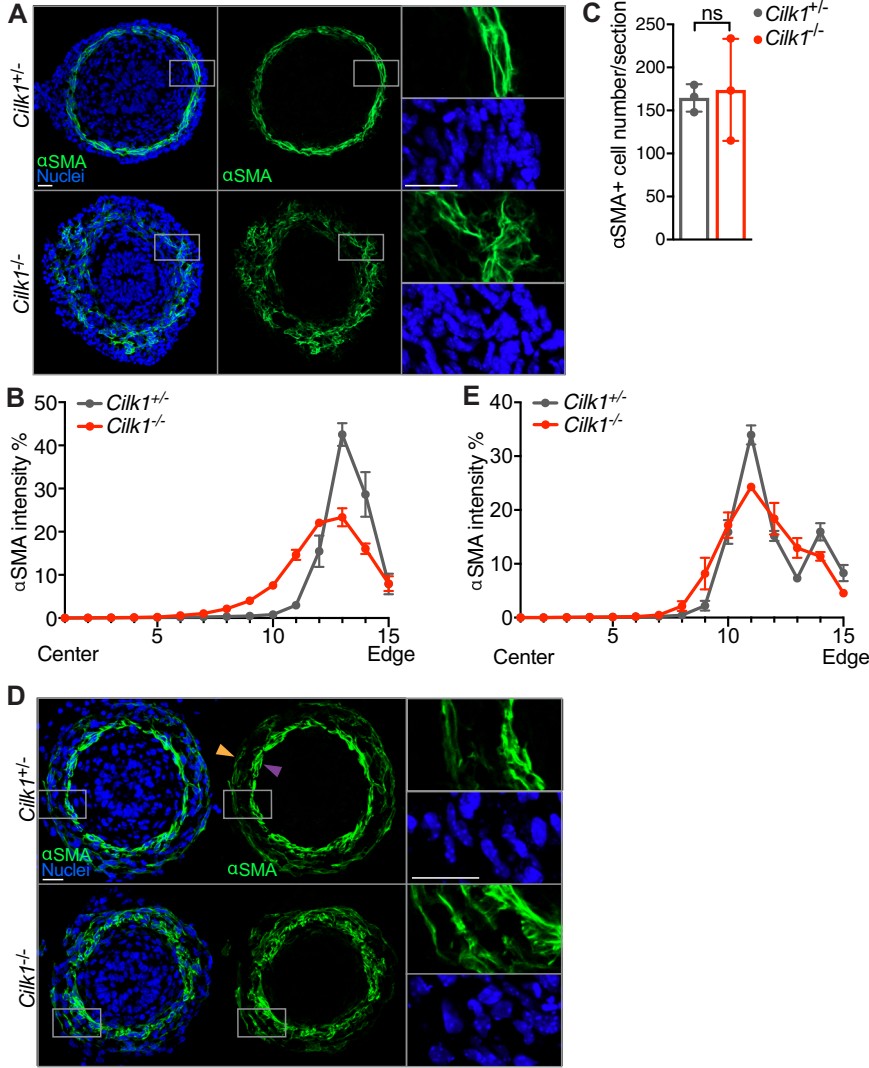

**Fig. 2 The circumferential smooth muscle cell layer is disrupted in the CILK1-deficient gut prior to length shortening. A** Immunofluorescence staining for smooth muscle (αSMA, green) and nuclei (Hoechst, blue) in E13.5 *Cilk1*[+/−] and *Cilk1*[−/−] intestines. **B** The radial distribution of αSMA staining intensity from the center of the section to the outer section edge of E13.5 *Cilk1*[+/−] (*n* = 4) and *Cilk1*[−/−] (*n* = 4) intestines, subdivided into 15 ring bins. **C** The number of αSMA-positive cells per section from E13.5 *Cilk1*[+/−] (*n* = 3) and *Cilk1*[−/−] (*n* = 3) intestines. For each embryonic intestine, the number of αSMA-positive cells was averaged from 3–8 stained sections. Values are presented as means ± SD. ns, *p* > 0.05 by two-sided unpaired *t* test. **D** Immunofluorescence staining for smooth muscle (αSMA, green) and nuclei (Hoechst, blue) in E14.5 *Cilk1*[+/−] and *Cilk1*[−/−] esophagi. Upper panel, arrowheads indicate two layers of esophageal muscularis: lamina muscularis interna (purple) and the lamina muscularis externa (orange). **E** The radial distribution of αSMA staining intensity from the center of the section to the outer section edge of E14.5 *Cilk1*[+/−] (*n* = 3) and *Cilk1*[−/−] (*n* = 3) esophagi, subdivided into 15 ring bins. **A**, **D** right, higher magnifications of boxed regions. Scale bars, 25 μm. **B**, **E** The αSMA intensity fraction in each bin was averaged from 3–8 sections. Values are presented as means ± SEM. *X* axis, bin number. Source data are provided in the Source Data file.

GFP, indicating the *Myh11*[Cre-EGFP] is active in most smooth muscle cells (Supplementary Fig. 8A, upper panel). As expected, GFP expressing cells were mostly absent in *Myh11*[Cre-EGFP] *R26*[DTA176/+] embryos (Supplementary Fig. 8A, lower panel). The number of αSMA-positive cells in *Myh11*[Cre-EGFP] *R26*[DTA176/+] embryos was reduced to 63% of controls (Supplementary Fig. 8B).

Partially ablating the smooth muscle of *Myh11*[Cre-EGFP] *R26*[DTA176/+] embryos did not grossly alter embryonic development at E15.5 (Supplementary Fig. 8C). However, *Myh11*[Cre-EGFP] *R26*[DTA176/+] embryos exhibited shortened small intestines (Fig. 3A, B). Thus, the smooth muscle is critical for elongation, indicating that ciliary HH signaling patterns the smooth muscle to drive elongation.

To understand how smooth muscle promotes elongation, we explored how the circumferential smooth muscle contributes to

mechanical characteristics. Tubular organs, including the gut, exhibit a force called residual stress even in the absence of external loads[52]. We hypothesized that the smooth muscle is a source of circumferential residual stress. To test this hypothesis, we measured the opening angles (*θ*) resulting from longitudinally opening gut segments (Fig. 3C), indicators of tensile residual stress[53–55].

To assay whether intestinal residual stress is produced by smooth muscle, we measured the opening angle of intestinal segments of control and *Myh11*[Cre-EGFP] *R26*[DTA176/+] embryos. The opening angles were reduced in *Myh11*[Cre-EGFP] *R26*[DTA176/+] embryos (94.1° ± 8.0) as compared to controls (wild type 117.4° ± 10.4, *Myh11*[Cre] 114.4° ± 6.6, *R26*[DTA176/+] 118.6° ± 4.2) (Fig. 3D). Therefore, smooth muscle contributes to circumferential residual stress. The residual stress measured by opening

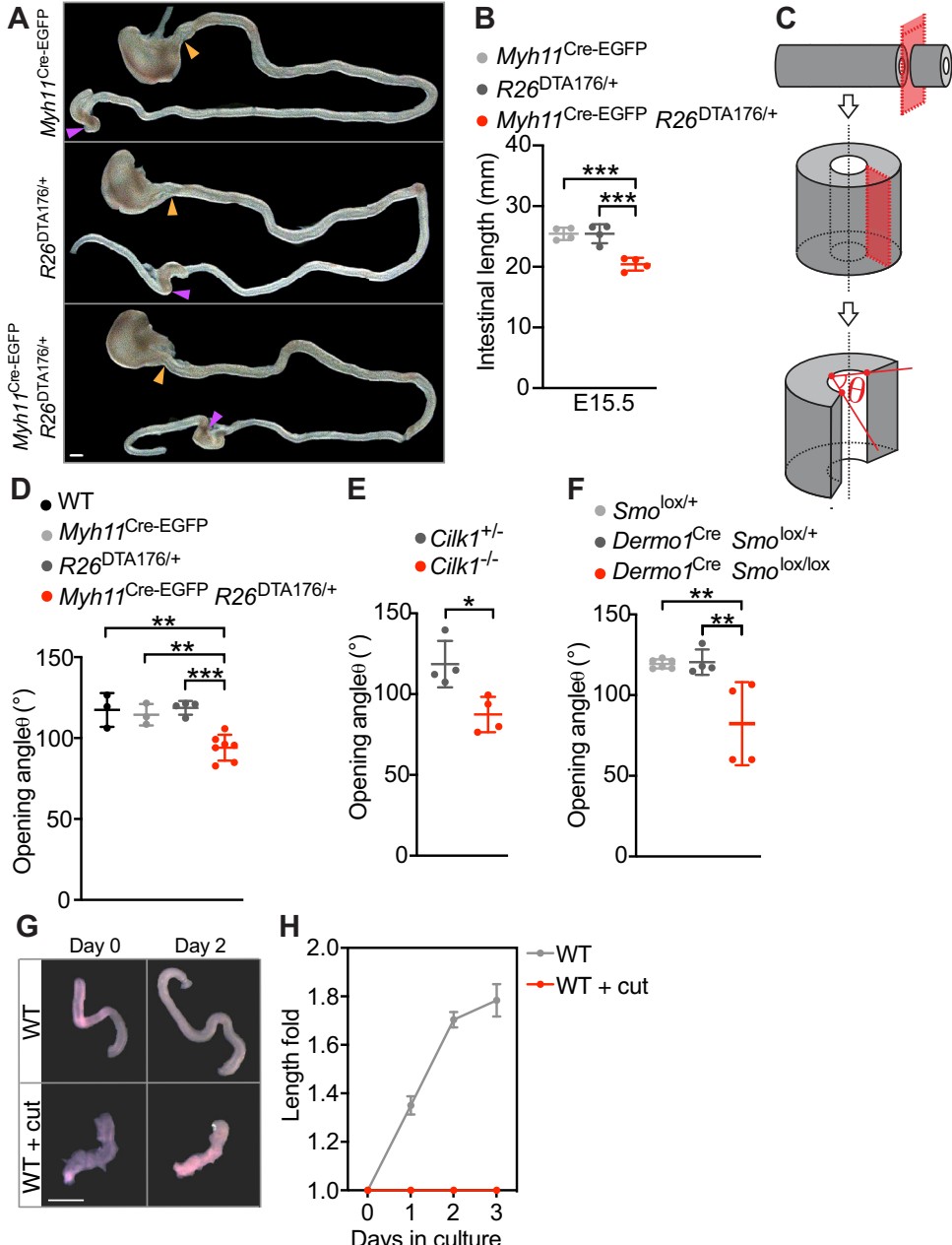

**Fig. 3 The circumferential smooth muscle creates residual stress to promote gut elongation. A** Intestines from E15.5 control ($Myh11^{Cre-EGFP}$ and $R26^{DTA176/+}$) and $Myh11^{Cre-EGFP}$ $R26^{DTA176/+}$ embryos. Scale bar, 1 mm. The length of the small intestine was measured from the end of the stomach (orange arrowheads) to the beginning of cecum (purple arrowheads). **B** Lengths of E15.5 control ($Myh11^{Cre-EGFP}$ $n = 4$ and $R26^{DTA176/+}$ $n = 4$) and $Myh11^{Cre-EGFP}$ $R26^{DTA176/+}$ ($n = 4$) small intestines. Each point in the scatter plots represents the value from an individual embryo. Values are presented as means ± SD. *** adjusted $P = 0.0007$ by ordinary one-way ANOVA Tukey's multiple comparisons test. **C** Schematic of method for measuring the opening angle to assess residual stress. We isolated approximately 1 mm long sections of E13.5 small intestine, cut the intestinal wall axially and measured the opening angle ($\theta$), with the vertex at the midpoint of the inner epithelial surface and the rays intersecting the innermost edges of the cut wall. **D** The opening angles of intestinal segments from E13.5 control (wild type, $n = 3$, $Myh11^{Cre-EGFP}$, $n = 3$ and $R26^{DTA176/WT}$, $n = 4$) and $Myh11^{Cre-EGFP}$ $R26^{DTA176/+}$ ($n = 7$) embryos. Adjusted $P$ values are 0.0018 for $Myh11^{Cre-EGFP}$ $R26^{DTA176/+}$ vs WT, 0.0054 for $Myh11^{Cre-EGFP}$ $R26^{DTA176/+}$ vs $Myh11^{Cre-EGFP}$, and 0.005 for $Myh11^{Cre-EGFP}$ $R26^{DTA176/+}$ vs $R26^{DTA176/WT}$. **E** The opening angles of intestinal segments from E13.5 $Cilk1^{+/-}$ ($n = 4$) and $Cilk1^{-/-}$ ($n = 4$) embryos. *$p = 0.0138$ by two-sided unpaired $t$ test. **F** The opening angles of intestinal segments from E13.5 ($Smo^{lox/+}$, $n = 6$ and $Dermo1^{Cre}$ $Smo^{lox/+}$, $n = 4$) and $Dermo1^{Cre}$ $Smo^{lox/lox}$ ($n = 4$) embryos. Adjusted $P$ values are 0.0038 for $Dermo1^{Cre}$ $Smo^{lox/lox}$ vs $Smo^{lox/+}$ and 0.0058 for $Dermo1^{Cre}$ $Smo^{lox/lox}$ vs $Dermo1^{Cre}$ $Smo^{lox/+}$. **G** Representative E13.5 wild-type intestine segments opened axially (WT + cut) or not before (Day 0) and after culture in Matrigel for 2 days (Day 2). Scale bar, 1 mm. **H** Length fold change of the cultured intestine segments (WT: $n = 3$; WT + cut: $n = 3$) for indicated days. **D**–**F** For each intestine, we calculated the median angle of 5–15 segments. **D**–**F** and **H** values are presented as means ± SD. *$p < 0.05$; **$p < 0.01$; ***$p < 0.001$ by ordinary one-way ANOVA Sidak's multiple comparisons test, except in **E**. Source data are provided in the Source Data file.

angles in the absence of supplemented extracellular calcium at room temperature likely reflect forces apart from any exerted by muscle contraction.

As partial ablation of the smooth muscle decreased residual stress in the intestine, we predicted that altered cilia function or mesenchymal HH signaling, both of which disrupt smooth muscle patterning, should also decrease residual stress. To begin to test this prediction, we examined the opening angles of $Cilk1^{-/-}$ intestines and found that compromising cilia function by loss of CILK1 reduced the opening angle by $31.1° ± 9.0$ (Fig. 3E).

We also examined the opening angle of $Dermo1^{Cre} Smo^{lox/lox}$ intestines and found that disruption of mesenchymal HH signaling also reduced intestinal residual stress (Fig. 3F). Therefore, disruption of the smooth muscle by genetic ablation, impaired cilia, or decreased HH signaling, each reduces intestinal residual stress and compromises elongation. Together, these data raised the interesting possibility that residual stress produced by the circumferential smooth muscle promotes longitudinal growth.

To test this possibility, we established an in vitro culture system that supports intestinal elongation. Consistent with in vivo results, $Dermo1^{Cre} Cilk1^{lox/lox}$ intestines exhibited decreased elongation in vitro (Supplementary Fig. 9A, B). To test whether residual stress is required for intestinal elongation in vitro, we released residual stress by incising intestines longitudinally. Cultured longitudinally incised intestines continued to proliferate and displayed no increase in apoptosis (Supplementary Fig. 9C, D). However, they failed to elongate (Fig. 3G, H), suggesting that residual stress in the intestine is important for its elongation.

Active contractions contribute to force produced by smooth muscle. In embryonic chicken intestine, the contractions of the circumferential smooth muscle promote anisotropic growth[56]. To assess mouse smooth muscle contractility in developing mouse intestine, we stained for phosphorylated Myosin Light Chain (phospho-MLC), a component of activated myosin[57]. Smooth muscle was positive for phospho-MLC at E18.5, but not at E13.5 (Supplementary Fig. 10A). Two-day treatment of Y-27632, a Rho-associated coiled-coil containing protein kinase (ROCK) inhibitor[58], did not alter the elongation of cultured mouse intestines (Supplementary Fig. 10B). As Y-27632 has been shown to inhibit muscle contraction at this concentration in multiple systems (but not confirmed here in the embryonic mouse gut)[59,60], our data suggest that smooth muscle contraction may not be essential for mouse intestinal elongation at E13.5. Thus, smooth muscle contraction may contribute to generating intestinal forces and elongation at other stages or in specific species.

**Smooth muscle activates YAP in the gut.** How does force promote the elongation of tubular organs? In Drosophila development, mechanical strain is interpreted by the Hippo pathway[61]. The primary effector of the Hippo pathway is the transcriptional co-activator YAP, which translocates to the nucleus upon activation. To assess whether the Hippo pathway also interprets mechanical stresses in the developing gut, we stained E13.5 intestine for YAP. Interestingly, the nuclear localization of YAP depended on radial location: inner mesenchymal cells between the epithelium and the smooth muscle (peri-epi-me cells, Supplementary Fig. 11A) displayed higher levels of nuclear YAP, and more peripheral smooth muscle cells possessed lower levels of nuclear YAP (Fig. 4A, top panel).

If YAP responds to force generated in the intestine by the circumferential smooth muscle, we predicted that disrupting the smooth muscle would decrease YAP nuclear levels. To test this prediction, we examined YAP localization in $Cilk1^{-/-}$ intestine. Interestingly, the levels of nuclear YAP were decreased in $Cilk1^{-/-}$

peri-epithelial stromal cells, indicating that CILK1 regulates YAP activity in the intestinal wall, directly or indirectly (Fig. 4A–C). Similarly, in $Cilk1^{-/-}$ esophagus, smooth muscle organization was disrupted, and nuclear YAP was reduced (Fig. 4D and S11B). Consistent with the reduced YAP protein levels, the expression of YAP-regulated proliferative genes ($Edn1$[62], $Ano5$[63], and $Ccl11$[64]) was decreased in $Cilk1^{-/-}$ intestine (Fig. 4E).

Similarly, the levels of nuclear YAP were decreased in $Dermo1^{Cre} Smo^{lox/lox}$ peri-epithelial mesenchymal cells, revealing that mesenchymal HH signaling also promotes YAP nuclear levels (Supplementary Fig. 11C, D). Moreover, the levels of nuclear YAP were also decreased in $Myh11^{Cre-EGFP} R26^{DTA176/+}$ mesenchymal cells (Supplementary Fig. 11E, F). Together these results demonstrate that intestinal smooth muscle is critical for YAP activation. As mechanical strain activates YAP to drive cell proliferation in mammalian cells[65], we hypothesized that YAP translates the force generated by the smooth muscle into growth.

**Mesenchymal Hippo signaling promotes elongation.** To test whether Hippo signaling participates in elongation, we generated $Dermo1^{Cre} Yap^{lox/lox} Taz^{lox/+}$ embryos. $Dermo1^{Cre} Yap^{lox/lox} Taz^{lox/+}$ embryos were grossly normal and exhibited no alteration in body size (Supplementary Fig. 12A). $Dermo1^{Cre} Yap^{lox/+} Taz^{lox/lox}$ intestines were 10% shorter than controls (Fig. 5A, B), indicating that TAZ makes a more minor contribution to elongation. $Dermo1^{Cre} Yap^{lox/lox} Taz^{lox/+}$ intestines were 40% shorter than those of controls (Fig. 5A, B), demonstrating that mesenchymal YAP activity is critical for elongation.

To examine whether mesenchymal YAP, like CILK1, promotes proliferation, we quantified EdU labeling of $Dermo1^{Cre} Yap^{lox/lox} Taz^{lox/+}$ and control intestines at E12.5. The percentage of EdU-labeled cells was reduced in $Dermo1^{Cre} Yap^{lox/lox} Taz^{lox/+}$ intestines both in the mesenchyme and epithelium (Fig. 5C, D), indicating that the proliferation is promoted by both CILK1 and YAP. The expression of pro-proliferative YAP-genes $Edn1$, $Ano5$ and $Ccl11$ is also decreased in $Dermo1^{Cre} Yap^{lox/lox} Taz^{lox/+}$ intestine as in $Cilk1^{-/-}$ intestine (Figs. 4E and 5E). E13.5 $Dermo1^{Cre} Yap^{lox/lox} Taz^{lox/+}$ intestines displayed no alteration in smooth muscle development (Fig. 5F and S12B-C). These findings, that YAP is dispensable for smooth muscle generation, but critical for proliferation, is consistent with the hypothesis that mesenchymal YAP is downstream of the circumferential smooth muscle to promote growth.

**Cilia drive elongation through mechanical activation of YAP.** To test whether YAP mediates the elongation promoted by ciliary signaling, we treated cultured embryonic intestines with XMU-MP-1, a selective inhibitor of YAP upstream kinases MST1/2. Inhibiting MST1/2 kinase with XMU-MP-1 activates YAP[65]. As shown before (Supplementary Fig. 9A, B), the elongation of cilia-mutated intestine was reduced in culture comparing to controls (Fig. 6A, B, DMSO). Notably, XMU-MP-1 treatment restored the elongation of the $Cilk^{-/-}$ intestine (Fig. 6A, B, XMU-MP-1). This rescue indicates that the suppression of YAP activities in cilia-mutated intestine causes the intestinal shortening.

To test how mechanical influences intestinal elongation, we cultured the embryonic intestines in different percentages of Matrigel. From 50% to 90%, the stiffness of Matrigel increases linearly[66]. Increased extracellular matrix stiffness promotes nuclear translocation of YAP[67]. Consistent with these observations, the nuclear levels of YAP were elevated in intestines cultured in 90% Matrigel (Fig. 6C, D). Interestingly, the elongation of $Cilk^{-/-}$ intestine was partially restored when cultured in 90% Matrigel (Fig. 6E). Together, our data suggest

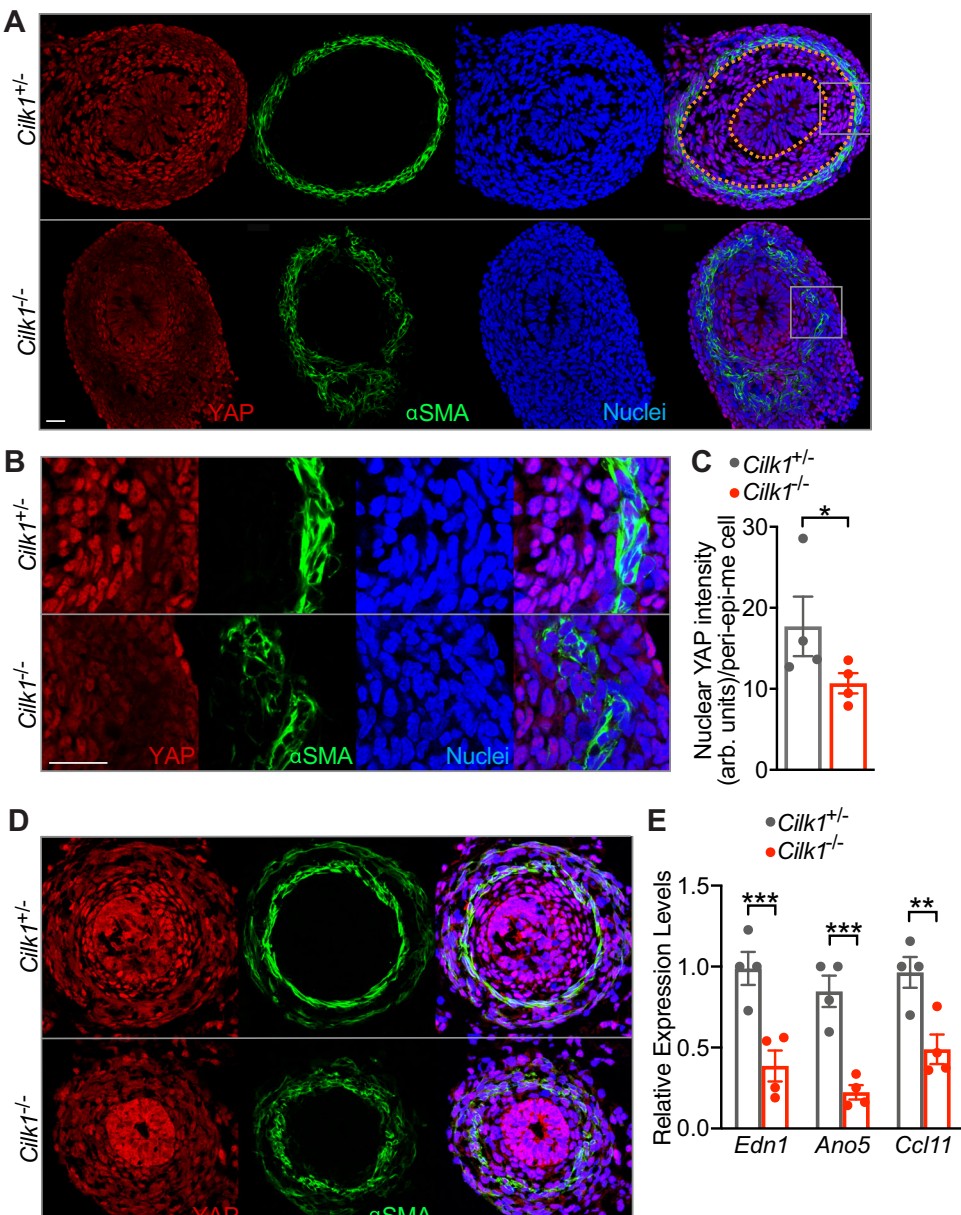

**Fig. 4 YAP nuclear levels and activity depend on CILK1. A** Immunofluorescence staining of E13.5 *Cilk1⁺/⁻* and *Cilk1⁻/⁻* intestines for YAP (red), smooth muscle (αSMA, green), and nuclei (Hoechst, blue). Scale bar, 25 μm. Orange dotted lines outline the epithelium and smooth muscle cell boundaries, in between are the peri-epithelial mesenchymal cells. **B** Magnifications of the boxed areas from **A**. Scale bar, 25 μm. **C** Nuclear YAP intensities of peri-epithelial mesenchymal (peri-epi-me) cells in E13.5 *Cilk1⁺/⁻* (*n* = 4) and *Cilk1⁻/⁻* (*n* = 4) intestines. For each intestine, the nuclear YAP intensity per cell was averaged from three sections. Values are presented as means ± SEM. *$p$ = 0.0176 by two-sided ratio paired $t$ test. **D** Immunofluorescence staining of E14.5 *Cilk1⁺/⁻* and *Cilk1⁻/⁻* esophagi for YAP (red), smooth muscle (αSMA, green) and nuclei (Hoechst, blue). Scale bar, 25 μm. **E** Relative levels of *End1*, *Ano5*, and *Ccl11* mRNA of E13.5 control *Cilk1⁺/⁻* (*n* = 4) and *Cilk1⁻/⁻* (*n* = 4) intestines. Values are presented as means ± SEM. **$p$ < 0.01 and ***$p$ < 0.001 by two-sided unpaired $t$ test (0.0051 for *Edn1*, 0.0011 for *Ano5* and 0.0114 for *Ccl11*). Source data are provided in the Source Data file.

that the elongation of developing tubular organs is driven by YAP mechanically activated by cilia-patterned smooth muscle.

## Discussion

Our results reveal a mechanism of tubular organ elongation by which intercellular signaling events create mechanical forces that are then translated into longitudinal growth. First, HH signals produced by epithelial cells are interpreted by the surrounding mesenchymal cells in a CILK1- and cilium-dependent manner. Second, ciliary HH signaling in the mesenchyme patterns the developing circumferential smooth muscle. Third, the smooth muscle contributes to residual stresses that exert mechanical forces on the mesenchymal cells. Fourth, these mesenchymal cells sense the smooth muscle-dependent forces, triggering nuclear accumulation of the Hippo pathway effector YAP. Fifth, nuclear YAP promotes cell proliferation, resulting in elongation. Thus, for the developing gut to elongate, morphogen-mediated patterning generates mechanical forces that drive YAP-promoted proliferation. As these findings apply to both the small intestine and the esophagus, we propose that the mechanical influences generated by ciliary signaling-patterned smooth muscle may be a general mechanism driving the elongation of tubular organs.

Our study provides a mechanistic explanation of how HH signaling regulates the intestinal length. The role of

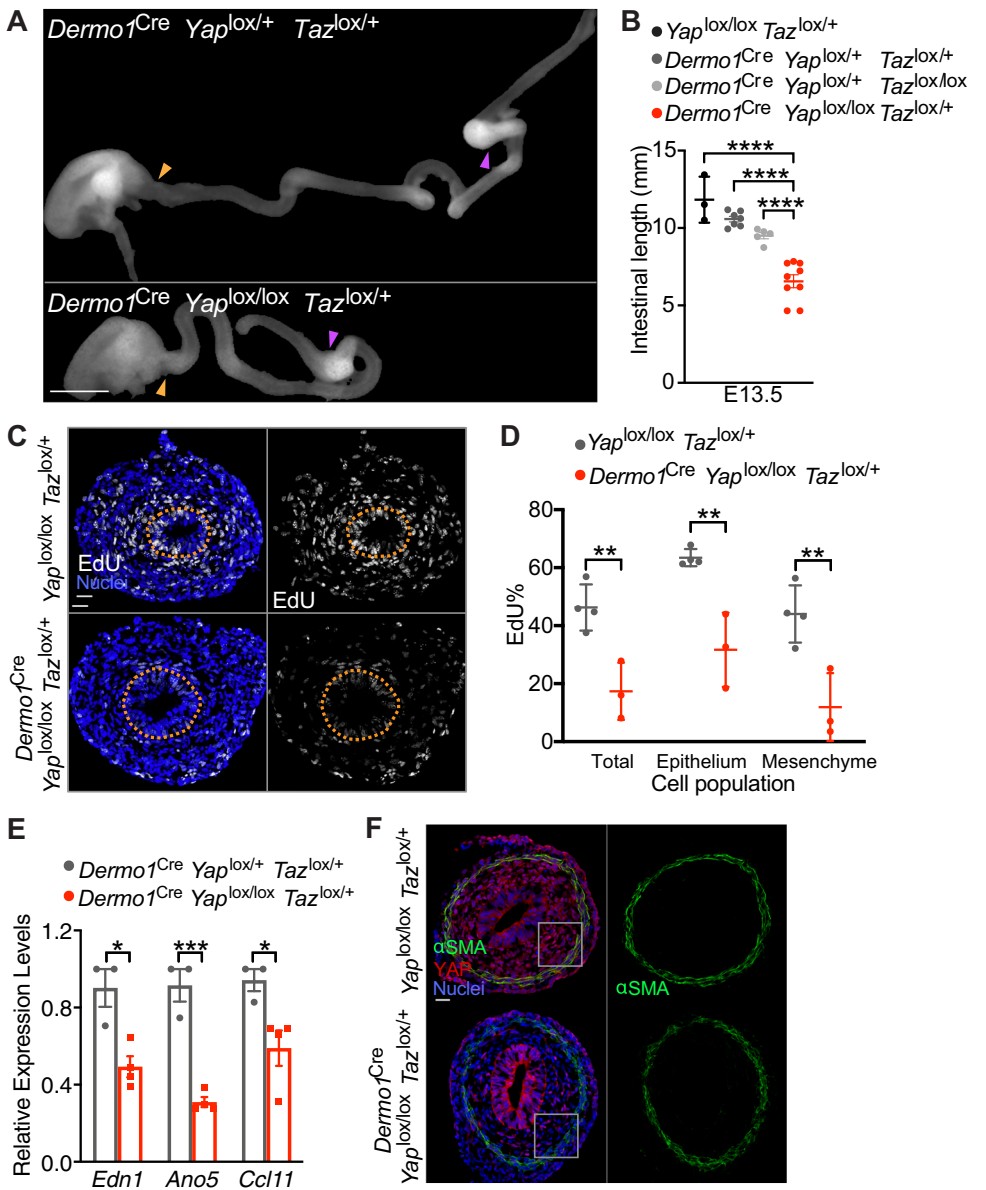

**Fig. 5 YAP functions in the mesenchyme to promote gut elongation and cell proliferation. A** Photos of E13.5 *Dermo1*^Cre *Yap*^lox/+ *Taz*^lox/+ and *Dermo1*^Cre *Yap*^lox/lox *Taz*^ox/+ intestines. Scale bar, 1 mm. The length of the small intestine was measured from the caudal stomach (orange arrowheads) to the rostral cecum (purple arrowheads). **B** Lengths of E13.5 in *Yap*^lox/lox *Taz*^lox/+ (*n* = 3), *Dermo1*^Cre *Yap*^lox/+ *Taz*^lox/lox (*n* = 7), *Dermo1*^Cre *Yap*^lox/+ *Taz*^lox/+ (*n* = 5) and *Dermo1*^Cre *Yap*^lox/lox *Taz*^ox/+ (*n* = 9) intestines. Horizontal bars indicate means ± SD. ****$p < 0.0001$ by one-way ANOVA Sidak's multiple comparisons test. **C** Immunofluorescence staining of E12.5 *Yap*^lox/lox *Taz*^lox/+ and *Dermo1*^Cre *Yap*^lox/lox *Taz*^lox/+ intestines for EdU (white) and nuclei (Hoechst, blue). Scale bar, 25 μm. Orange dotted lines outline the epithelium. **D** The percentage of EdU-positive cells in the E12.5 whole intestine, the epithelial cells or the mesenchymal cells in *Yap*^lox/lox *Taz*^lox/+ (*n* = 4) and *Dermo1*^Cre *Yap*^lox/lox *Taz*^lox/+ (*n* = 3). For each intestine, we averaged the percentages in the indicated cell population from three stained sections. Horizontal bars indicate means ± SD. **$p < 0.01$ by two-way ANOVA Sidak's multiple comparisons test (adjusted *P* are 0.0031 for Total, 0.0014 for Epithelium and 0.0013 for Mesenchyme). **E** Relative levels of *End1, Ano5,* and *Ccl11* mRNA in E13.5 control (*Dermo1*^Cre *Yap*^lox/+ *Taz*^lox/+, *n* = 3) and *Dermo1*^Cre *Yap*^lox/lox *Taz*^lox/+ (*n* = 4) intestines. Values are presented as means ± SEM. *$p < 0.05$ and ***$p < 0.001$ by two-sided unpaired *t* test (0.0114 for *Edn1*, 0.0005 for *Ano5* and 0.0311 for *Ccl11*). **F** Immunofluorescence staining of E13.5 *Yap*^lox/lox *Taz*^lox/+ and *Dermo1*^Cre *Yap*^lox/lox *Taz*^lox/+ intestines for YAP (red), smooth muscle (αSMA, green) and nuclei (Hoechst, blue). Higher magnifications of boxed regions are depicted in Supplementary Fig. 12B. Scale bar, 25 μm. Source data are provided in the Source Data file.

HH signaling in intestinal growth is therefore distinct from its roles in tissues such as the cerebellum and skin, where it directly induces the expression of drivers of the cell cycle[68,69]. Instead, the intestine represents a different model of how HH signaling functions in organ development: HH signaling patterns the gut, and this patterning, acting via mechanical influences interpreted by the Hippo pathway, promotes proliferation.

In support of the causal relationship between smooth muscle-generated forces and elongation, disruption of circumferential smooth muscle by any of three independent means (disrupting ciliary signaling, disrupting HH signaling, or genetic ablation of smooth muscle cells) each reduces elongation. Loss of ciliary proteins such as CILK alters the organization of the smooth muscle, whereas loss of SMO decreases the amount of smooth muscle. We propose that, as in other tissues, the different effects

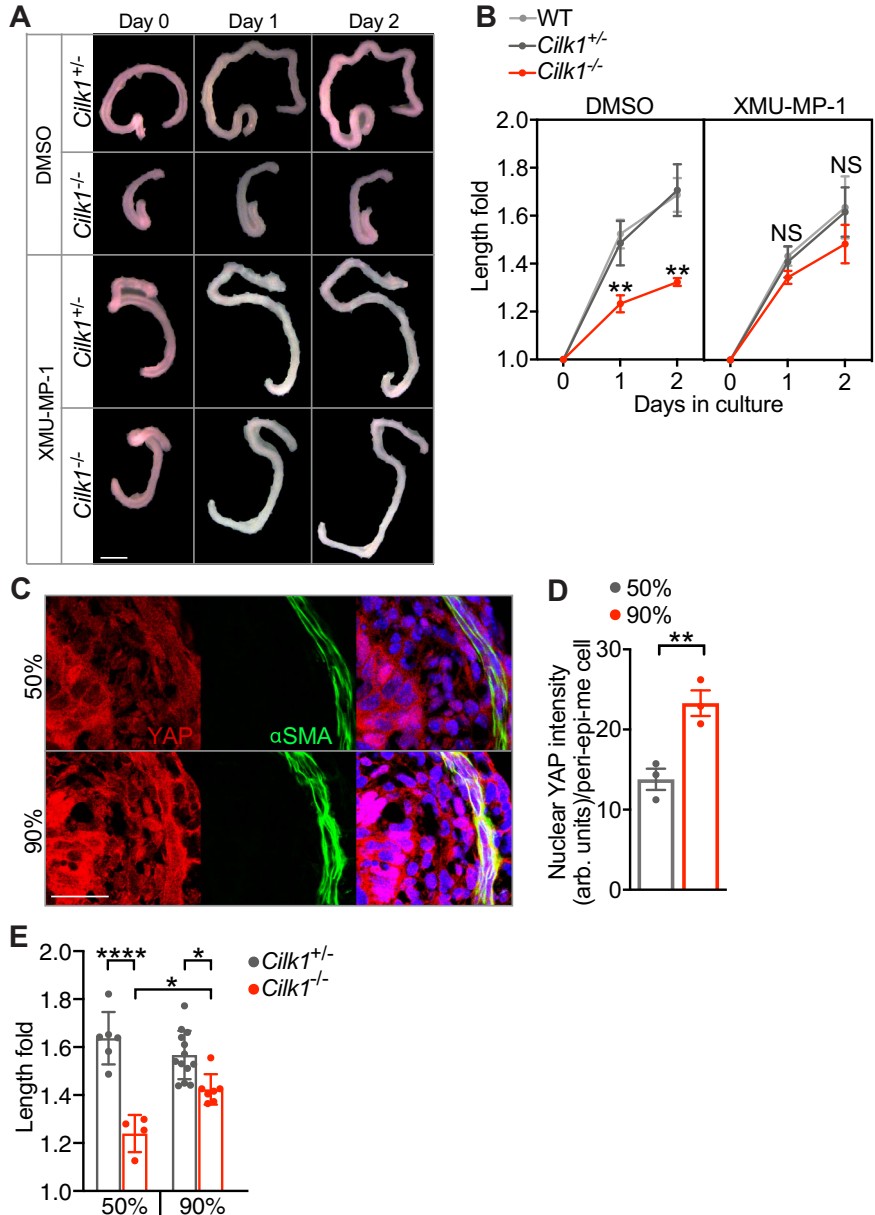

**Fig. 6 Modulating the Hippo pathway restores elongation of *Cilk1*⁻/⁻ intestine. A** Representative E13.5 intestines of indicated genotypes before (Day 0) and after culture in Matrigel with DMSO or XMU-MP-1 for indicated days. Scale bar, 1 mm. **B** Fold change of intestinal length after culture in Matrigel with DMSO (WT, $n = 6$; *Cilk1*⁺/⁻, $n = 5$; *Cilk1*⁻/⁻, $n = 4$) or XMU-MP-1 (WT, $n = 3$; *Cilk1*⁺/⁻, $n = 4$; *Cilk1*⁻/⁻, $n = 6$) for indicated days. The values are presented as means ± SD. ns $p > 0.05$, **$p < 0.01$ by two-way ANOVA Sidak's multiple comparisons test (adjusted $P$ values are 0.0046 for day 1 and 0.0025 for day 2 between *Cilk1*⁺/⁻ DMSO and *Cilk1*⁻/⁻ DMSO). **C** Immunofluorescence staining of E13.5 wild-type intestine cultured for one day in 50% or 90% Matrigel for YAP (red), smooth muscle (αSMA, green) and nuclei (Hoechst, blue). Scale bars, 25 μm. **D** Nuclear YAP intensities of peri-epithelial mesenchymal cells in **C**. $n = 3$ for each group. For each intestine, the nuclear YAP intensity per cell was averaged from three sections. Values are presented as means ± SEM. **$p = 0.035$ by two-sided paired $t$ test. **E** Fold change of intestinal length after culture in 50% (*Cilk1*⁺/⁻, $n = 6$; *Cilk1*⁻/⁻, $n = 4$ or 90% (*Cilk1*⁺/⁻, $n = 13$; *Cilk1*⁻/⁻, $n = 7$) Matrigel for two days. The values are presented as means ± SD. *$p < 0.05$, ****$p < 0.0001$ by one-way ANOVA Turkey's multiple comparisons test. Adjusted $P$ value are: 0.0194 for *Cilk1*⁻/⁻ 50% vs *Cilk1*⁻/⁻ 90% and 0.0145 for *Cilk1*⁺/⁻ 90% vs *Cilk1*⁻/⁻ 90%. Source data are provided in the Source Data file.

of loss of ciliary proteins and SMO on smooth muscle formation are attributable to differential processing of GLI transcription factors, the effectors of HH signaling[21,70]. Importantly, altering the organization or the number of smooth muscle cells decreases force within the developing intestine and the nuclear levels of YAP, presaging the emergence of decreased intestinal elongation. As mesenchymal YAP is essential for elongation and activating YAP can rescue the shortening of cultured cilia-compromised intestines, we propose that smooth muscle within the gut wall

provides the mechanical growth cues which YAP interprets to drive elongation. This mechanism is likely to underlie the etiology of congenital short bowel in human ciliopathies that compromise HH signaling[71]. Prior to the development of the smooth muscle, intestinal elongation may be independent of mechanical forces or dependent on forces generated through other mechanisms, such as extracellular matrix stiffness or vitelline duct tension[72].

It remains unclear how forces in the intestine and YAP activation lead to anisotropic growth. One possibility is that

mechanical forces promote cell division oriented along the longitudinal axis, consistent with the decreased longitudinally-oriented mitoses in *Cilk1* mutant intestines. Another possibility is that squeezing by the smooth muscle directly leads to intestinal elongation[56]. Still other possibilities exist, such as centripetal forces driving cell migration along the orthogonal longitudinal axis. As inhibiting the function of either CILK1 or YAP in the mesenchyme decreased proliferation in both the mesenchyme and the epithelium, there are likely to also be mechanisms for coordinating proliferation between the two cell types.

Many well studied cellular responses to mechanical forces are acute. For example, the cytoskeleton responds to forces within seconds[73,74]. Developmental processes typically occur over much longer timescales. In Drosophila, mechanical cues can affect epithelial planar organization to drive elongation[75,76]. In the mammalian gut, our findings reveal that the smooth muscle-generated residual stress stores durable spatially-organized mechanical information. We propose that residual stress persists to enable tissue-level responses, such as gut elongation, over developmental timescales. Interestingly, mechanical cues can also be stored in extracellular matrix-based basement membrane as a stiffness gradient to drive elongation of the Drosophila follicle[77]. Similar to our findings, mechanical forces activates YAP and elongates the Drosophila follicle, although proliferation is not involved[61]. Static forces can activate YAP via deformation of the nucleus[78]. Whether circumferential force activates YAP via nuclear stretching will be an interesting subject of further investigation.

The effects of mechanical forces in vertebrate tissue growth have been most extensively documented in bone remodeling[79,80]. During development, our data suggest that mechanical forces, as in bone, promote growth, but these forces are internally generated by smooth muscle within the gut wall. Longitudinal forces transmitted by the vitelline duct also promote elongation of embryonic intestine in the chick embryo[72]. The relative growth between the gut and its connecting tissues generates forces that shape the looping of gut tube across species[81], suggesting that organ elongation is regulated by both internal and external forces.

As our data indicate that mechanical force can promote elongation of tubular organs during development, we speculate that activation of mechanotransducive signaling may also promote regeneration in adult tissues. This speculation is supported by emerging evidence showing that reversing the tissue mechanics associated with aging can rejuvenate the adult stem cells[82]. The adult intestine is one of the most proliferative organs, with human intestinal epithelial cells turning over every 2–5 days[83] driven by intestinal epithelial stem cells which demonstrate robust regenerative capacity also in vitro[84,85]. However, guts do not display compensatory lengthening postnatally, either after injury or congenital defects that cause shortening. Consequently, individuals with short bowel syndrome do not recover intestinal length over time, leading to chronic water and nutrient absorption deficits[86]. Interestingly, application of longitudinal mechanical force can promote postnatal intestinal growth in animal models[2]. Our findings raise the possibility that application of radial forces may be therapeutically useful for promoting postnatal elongation of tubular organs.

We established an in vitro culture system that models the elongation of murine embryonic intestine in a way that recapitulates in vivo development. This culture system allows the explanted intestine to elongate and pharmacological rescue of the genetic phenotypes. We found that increasing the stiffness of the extracellular matrix increased YAP activity and promoted elongation of *Cilk1* mutant intestines. The stiffness of the matrix may compensate for the reduced force imposed by *Cilk1* mutant intestines. We propose that this in vitro system will be a valuable tool to further define the determinants of intestinal elongation.

To conclude, we uncovered an integrated molecular mechanism by which morphogen signaling patterns a tissue to create the mechanical cues that regulate organ size. During the development of the tubular organs, the activation of ciliary signaling in the mesenchyme patterns the wrapping smooth muscle layer, which generates intrinsic forces that activates the mechanotransducive effector YAP. Active YAP promotes proliferation resulting in growth. This cooperation between biochemical and mechanical signals to regulate organ elongation illustrated here may represent a general mechanism to control growth of tubular organs.

## Methods

**Mouse lines**. All mouse experiments were approved by the Institutional Animal Care and Use Committee of the University of Helsinki (license # ESAVI/3881/04.10.07/2015 and KEK15-001) and the University of California, San Francisco (UCSF, protocol # AN178683).

The frozen sperm heterozygous for an *Cilk1* mutant allele (also called *Ick*tm1a(KOMP)Mbp) were generated by the National Institutes of Health (NIH) Knockout Mouse Project (KOMP) and obtained from the KOMP Repository (www.komp.org). The mutant mouse line was recovered using in vitro fertilization by the Transgenic Core Facility of Biocenter Oulu at the University of Oulu and transferred to the Laboratory Animal Center at the University of Helsinki. In this mouse line, the *Cilk1* locus was modified by homologous recombination using a targeting vector PRPGS00177_A_B02 designed to insert a cassette expressing *lacZ* with polyA signal between exons 5 and 6, resulting in a predicted loss of function allele (Supplementary Fig. 1A, *Cilk1*-). *Cilk1*+/- mice were crossed to *Tg(CAG-Flpe)2Arte* transgenic mice ubiquitously expressing FlPe recombinase to generate the conditional *Cilk1* allele with loxP sites flanking exon 6 (Supplementary Fig. 1A, *Cilk1*lox).

To generate *Tctn3* loss of function allele (Supplementary Fig. 2A, *Tctn3*-), the targeting vector PRPGS00066_A_C10 was obtained from KOMP, linearized by restriction digestion with *Asi*SI (NEB), and electroporated into E14 mouse embryonic stem cells (ESC), followed by culture in G418-containing selection media. Homologous recombination was identified by PCR and a targeted clone was injected into blastocytes by the Transgenic/Gene-targeting Core Facility at the Gladstone Institute of UCSF.

To generate the *Ift88* null allele (*Ift88*-), *Ift88*lox/lox mice were crossed to *Tg(ACTB-cre)2Mrt* transgenic mice expressing Cre under the control of a human *beta-actin* promoter.

The source and genetic background information of the mouse lines used in this study are detailed in Reporting summary.

**Isolation of the small intestine and esophagus**. Uteruses of euthanized pregnant female mice were isolated and placed in DPBS (Dulbecco's Phosphate Buffered Saline, without calcium or magnesium). The embryos were removed from the uterus and transferred to a Petri dish containing DPBS on ice. To isolate intestine and esophagus, embryos were transferred individually to another Petri dish filled with DPBS and imaged using a Zeiss Discovery V12 stereo microscope. The digestive tract including stomach, small intestine, colon, and cecum was separated from adjoining tissues and removed en bloc. Incisions in the stomach and cecum isolated the intestine, and the intestinal mesentery removed.

To isolate the esophagus, the heart and lung were gently removed to expose the esophageal-gastric junction. The esophagus was followed up from the stomach to pharynx to separate from the surrounding trachea, vasculature, and mesentery.

**Intestinal and esophageal length measurement**. The straightened intestine and esophagus, together with a ruler, were imaged using Zeiss Discovery V12 stereo microscope. Curves in the intestine and esophagus were accounted for by measuring the length using the ImageJ segmented line tool. Figures shown were prepared using Photoshop (Adobe) to remove background for a clear view of the tissue.

**Matrigel culture of embryonic intestine**. The culture medium was made of BGJb medium with L-Ascorbic acid for a final concentration of 0.1 mg/mL[87] and penicillin/streptomycin solution for a final concentration of 100 U penicillin/mL. Matrigel were thawed on ice and mixed with the culture medium at 1:1 for 50% or 9:1 for 90% final Matrigel concentration. E13.5 intestines were isolated as described and transferred into microcentrifuge tubes with 50% or 90% Matrigel as indicated in the figures. 40 μl of the Matrigel mix containing the tissue was aspirated and added to 24-well plate. After 20 min in 37 °C incubator, 200 μl of the culture medium was carefully added to surround the solidified Matrigel drop, followed by photo as Day 0. The culture medium was changed every day. For drug treatments, XMU-MP-1 was dissolved in DMSO and added to 50% Matrigel and culture medium for a final concentration of 500 nM. Y-27632 was applied at 10 μM. The culture medium was added with fresh drug and changed daily.

**Cryosectioning**. Embryonic intestines and esophagi were fixed in 4% paraformaldehyde (PFA) for 1 h to overnight at 4 °C. The fixed tissue was washed with DPBS once then submerged in 30% sucrose in DPBS. After overnight incubation at 4 °C, the tissue was transferred to a cryomold (4730, Tissue-Tek) and embedded in O.C.T. compound (4583, Tissue-Tek) on dry ice. The frozen blocks were stored at -80 °C until sectioning. 7 μm sections were cut at -20 °C using a cryostat (Leica CM3050 S) and mounted directly on Super-frost Plus microscope slides (12-550-15, Fisher Scientific).

**Immunofluorescence staining**. Immunostaining was performed on 7 μm O.C.T. frozen sections. The sections were post-fixed in 4% PFA for 10 min at room temperature (RT) and washed three times with DPBS. After the washes, each slide was placed on a slide rack (Thermo Scientific Shandon Sequenza) and permeabilized with 200 μl of DPBS-0.5% Triton for 20 min, followed by blocking with 200 μl of 10% donkey serum in DPBS-0.25% Triton for 1 h. After blocking, 200 μl of primary antibody was applied. After overnight incubation at RT, each slide was washed three times with 1 ml DPBS-0.25% Triton and then incubated at RT with secondary antibody for 2 h in the dark. Washing was repeated and 200 μl of 1 mg/ml Hoechst H33342 was applied for 5 min prior to the final wash, after which a cover glass (22266882, Fisher Scientific) was mounted to the stained slide in the presence of Prolong Gold Antifade Reagent (P36930, Thermo Fisher Scientific). To visualize γ-tubulin, the sections were incubated in 1% SDS in DPBS for 5 min and washed with DPBS three times before blocking. Detailed information about antibodies used in this study is included in the Reporting summary.

**EdU labeling**. EdU (E10187, Thermo Fisher Scientific) and thymidine (T9250, Thermo Fisher Scientific) were weighed and reconstituted in DPBS at 2.5 mg/ml and 25 mg/ml, respectively. At E13.5, EdU solution was intraperitoneally (IP) injected into pregnant female mice at 50 mg/kg body weight. After 10 mins, thymidine solution was IP injected at 500 mg/kg to quench EdU incorporation. The injected females were euthanized by CO$_2$ inhalation followed by cervical dislocation. The embryonic intestines were fixed and sectioned, as described, then subjected to EdU staining after permeabilization. The EdU staining was performed according to the manufacturer's protocol (C10269, Thermo Fisher Scientific). Briefly, 200 μl of Click-iT reaction cocktail was incubated with each slide for 30 min in the dark. After two washes with 3% BSA in DPBS, Hoechst staining was performed as described in "Immunofluorescence Staining".

**Confocal imaging**. The stained sections were imaged using a Zeiss LSM 800 with 20X air or 63X oil objectives. The Zen version 2.5 blue edition software was used to acquire and export images.

**Opening angle assay**. E13.5 intestines were individually transferred to a Petri dish containing DPBS, examined using a Zeiss Discovery V12 stereo microscope and cut into approximately 1 mm cylinders. Using a 30 G needle, a single radial cut was made through one wall of the cylinder. A phase-contrast photomicrograph was taken immediately thereafter with a Zeiss Axiocam camera attached to the stereo microscope. Using the Angle Tool in ImageJ, the opening angle (θ) was measured between two lines intersecting the midpoint of the inner epithelial surface and extended to the inner edges of the cut wall, as shown in Fig. 3C.

**Epithelial-mesenchymal separation of E18.5 embryonic small intestine**. Small intestines from E18.5 embryos were cut longitudinally and incubated in ice-cold Cell Recovery Solution (354253, Corning) for 30–60 min with gentle shaking. The epithelium was gently separated from the mesenchyme with forceps. The isolated epithelium and mesenchyme were briefly washed in DPBS prior to lysing in TRIzol (15596026, Thermo Fisher Scientific) for RNA extraction.

**RNA extraction**. The tissue was incubated with 1 ml TRIzol in Lysing Matrix D (MP116913100, MP Biomedicals) and homogenized using TissueLyser II (85300, Qiagen) twice at a speed of 30 Hz for 30 s with an interval of 1 min. To each sample, 0.2 ml chloroform (372978, Sigma) was added and shaken again in a TissueLyser II (85300, Qiagen) at 30 Hz for 15 sec, followed by a 3 min incubation at RT and centrifugation at 12,000 x g for 15 min at 4 °C. The upper phase was transferred to an RNase-free tube with 0.5 ml isopropanol (I9516, Sigma) and inverted 10 times to precipitate RNA. After an incubation at RT for 10 min and centrifugation at 12,000 x g for 10 min at 4 °C, the RNA pellet was washed with 1 ml 75% ethanol and air dried. The pellet was then dissolved in 50 μl RNase-free water, and RNA concentration was measured using a spectrophotometer (Nanodrop 2000, Thermo Fischer Scientific).

**Real-Time quantitative PCR (RT-qPCR)**. 800 ng to 1 μg of total RNA was used as template to generate cDNA using the High Capacity cDNA Reverse Transcription Kit (4368814, Thermo Fisher Scientific). To avoid RNase contamination, RNase-free reagents and 20U RNase inhibitor (N8080119, Thermo Fisher Scientific) were used with each reaction. After thermal cycling according to the manufacture's manual, the cDNA was diluted to 20 ng/μl in UltraPure DNase/RNase-Free Distilled Water (10977023, Thermo Fisher Scientific). 1 μl of the diluted cDNA was

used as template in a 20 μl RT-qPCR reaction with KAPA SYBR FAST Universal 2X qPCR Master Mix (KK4602, Kapa Biosystems) and 200 nM forward and reverse primers (Sigma or Integrated DNA Technologies, sequences in Supplementary Table 1). The reactions were loaded into a 96-well plate and analyzed using the StepOnePlus Real-Time PCR system (Thermo Fisher Scientific) with an annealing temperature of 60 °C. The comparative threshold cycle method was used to analyze relative expression curves.

**Genotyping**. To genotype weaning-age pups, 100 μl of Buffer A (10 M NaOH, 500 mM EDTA) was added to mouse ear punches and incubated in PCR blots for 25 min at 99 °C and 5 min at 4 °C, followed by adding 100 μl of Buffer B (1 M Tris, pH 7.5). To genotype embryos, 100 μl of Buffer A was added to pieces of tail and incubated for 25 min at 99 °C and 5 min at 4 °C, followed by adding 100 μl of Buffer B with 0.5 mg/ml Proteinase K (03115828001, Roche) and incubated for 2 h at 55 °C, 10 min at 95 °C and 5 min at 4 °C. The DNA was stored at -20 °C until being assessed by PCR. To genotype alleles of $Myh11^{Cre-EGFP}$, $R26^{DTA176}$, $Smo^{lox}$, $YAP^{lox}$ and $TAZ^{lox}$, 1 μl of digested sample was added to a 20 μl reaction mixture with 200 nM forward and reverse primers and GoTaq G2 Green Master Mix (M7823, Promega Corporation) according to the manufacturer's instructions. The reactions were analyzed using a Biometra Trio Thermal Cycler (Analytik Jena) with an annealing temperature of 60 °C except for the $YAP^{lox}$ and $TAZ^{lox}$ reactions which were annealed at 62 °C. For other alleles, 0.25 μl of digested sample was added to a 5 μl reaction mixture with 200 nM forward and reverse primers and PowerUp SYBR Green Master Mix (A25742, Thermo Fisher Scientific) according to the manufacturer's instructions. The reactions were analyzed using an Applied Biosystems 7900HT Fast Real-Time PCR machine with an annealing temperature of 60 °C. Melt curve analysis was performed using SDS Software version 2.3 (Thermo Fisher Scientific). All the genotyping primers were synthesized by Integrated DNA Technologies and the sequences are included in Supplementary Table 1.

**Statistics and reproducibility**. All scatter plots, column, points & connecting line graphs and their associated statistical tests were generated using GraphPad Prism for macOS (version 8.1.2). The numbers of analyzed biological samples and the specific statistical analysis methods used for each experiment are described in the figure legends. The UCSF Clinical & Translational Science Institute biostatistics consultation service advised on the statistical tests and data presentation. Representative micrographs were drawn from experiments repeated at least three times as independent experiments.

**CellProfiler analysis of αSMA staining**. Quantification of αSMA staining was accomplished using a customized CellProfiler pipeline. The Hoechst and αSMA-stained images were imported to CellProfiler (version 3.1.8). To quantify the αSMA intensity distribution, a single object was created for the whole section area using the "Identify Primary Objects" and "Filter Objects" modules on Hoechst images. The center to the edge of the tissue section was divided into 15 areas of equal width, and the αSMA intensity within each area was measured using "FracAtD" in the "Measure Object Intensity Distribution" module on αSMA-stained images.

To quantify the αSMA-positive cell number, all the nuclei were identified by "Identify Primary Objects" with Hoechst staining as the input image. "Image Math" was used to multiply Hoechst by αSMA-stained images. After applied "Threshold", the multiplied images were converted to binary images, which were then used to mask the identified nuclei objects with a defined "Fraction of object that must overlap". The number of the resulted masked objects was used as the αSMA-positive cell number.

To quantify the αSMA area, αSMA signals were identified by "Identify Primary Objects" from αSMA-stained images. "Measure Image Area Occupied" was used to measure the area occupied by αSMA signals.

**CellProfiler analysis of YAP intensity**. The YAP intensity in the peri-epithelial stromal cells were done from staining of intestinal sections using customized CellProfiler pipelines. The Hoechst and YAP-stained images were imported to CellProfiler (version 3.1.8). The region occupied by peri-epithelial stromal cells were identified by "MaskImage" and "IdentifyObjectsManully" using merged images generated by "GrayToColor". The nuclei number were measured using "IdentifyPrimaryObjects" in the region and the nuclear YAP intensity by "MeasureImageIntensity" with the nuclei in the region as the input objects. In each intestinal section, the total intensity of nuclear YAP was divided by the number of nuclei in the region to generate nuclear YAP intensity per cell.

**Reporting summary**. Further information on research design is available in the Nature Research Reporting Summary linked to this article.

## Data availability
The data supporting the findings of this study are available in the main article and supplementary files. Source data are provided with this paper and are also available at [https://datadryad.org/stash/share/t9_ojoQ1CMESKEPYOsFXa0uPuK1N3ON5FU13QjYoqE8]. Source data are provided with this paper.

## Code availability

ImageJ macros and CellProfiler pipelines used for image analysis are available from the corresponding authors upon request.

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

## Acknowledgements

We are grateful to Elaine Carlson for help in generating *Tctn3* mutant alleles and E Yu for husbandry and genotyping of the mice used in this study. We thank Amy E. Shyer, Tyler R. Huycke, Wallace Marshall, Zev J. Gartner and Ophir D. Klein for thoughtful discussions. We also thank the members of the Reiter lab and the Makela lab for critical comments and suggestions on the study. This work was funded by NIH NIGMS R01GM095941, R01 AR054396, and NICHD R01HD089918 to J.F.R., NIH P30 DK090868 to K.E.M., and UCSF Program for Breakthrough Biomedical Research (PBBR) to K.E.M. and J.F.R.

## Author contributions

J.F.R. and Y.Y. conceived the study and designed the experiments, except for the observations in Figs. 1A, 1B, 1E, S1B, S1C, S1D, and S3, which were initially made by Y.Y. and P.P. with T.P.M. Y.Y. with K.E.M. and J.F.R made the findings in Fig. 1F, Supplementary Fig. 2B–F, and S4A-B. P.P. generated the data in Supplementary Fig. 3B–C and repeated the results in Figs. 1A, 1B, 1F, S1B, S4B independently. C.X. generated the data in Supplementary Fig. 4D. A.L.K. generated the *Tctn3* mutant mouse line. Y.Y. performed all the other experiments. J.F.R and Y.Y. wrote and prepared the manuscript with input from all authors.

## Competing interests

The authors declare no competing interests.
