## [Peer Review File · Nature Communications]

Ciliary Hedgehog signaling patterns the digestive system to generate mechanical forces driving elongationReviewers' Comments:

Reviewer #1:

Remarks to the Author:

Yang et al. provide evidence for a novel mechanism in which ciliary signaling elongates the mouse intestine and esophagus by activating YAP in circumferential smooth muscle cells. They show that ciliary and SMO mutants, which disrupt intestinal elongation, impact smooth muscle cell organization and proliferation, and that the primary requirement for cilia and SMO is in the mesenchymal cells surrounding the epithelial tubes. Defects in intestinal elongation are recapitulated by ablation of smooth muscle cells, further indicating that the smooth muscle is required for intestinal elongation. By making longitudinal cuts in the intestine, the authors show that the smooth muscle is a source of residual stress in the intestine at the time of elongation, and that this residual stress requires cilia and HH signaling. Using an in vitro culture system, the authors show that longitudinal cuts that reduce residual stress also prevent intestinal elongation, although these cuts appear to cause cell damage that could complicate the interpretation of these results.

The authors go on to show that ciliary mutants have reduced nuclear localization of YAP in peri-epithelial stromal cells and reduced expression of several YAP target genes. Intestines homozygous for YAP and heterozygous for TAZ in the mesenchyme have reduced elongation and decreased cell proliferation, indicating that YAP activity in the mesenchyme is required for elongation and that YAP is an important target of HH activity. They show using their cell culture system that activating YAP rescues intestinal elongation in a Cilk mutant intestine, indicating that reduced YAP activity is the cause of the intestinal elongation defects in cilia mutants.

This study describes several new genetic tools and provides valuable data on how tubes are organized and elongate. The results support the model that ciliary signaling elongates the intestine by directing cellular organization and YAP signaling in the circumferential smooth muscle layer. While the role of HH in intestinal elongation was known previously, the authors now link this signal to altered changes in YAP activity and show that YAP regulates intestinal elongation downstream of HH, linking two major signaling pathways to changes in tissue shape and mechanics. Because reduced intestinal elongation is associated with reduced proliferation and disrupted smooth muscle organization, the authors posit a mechanical feedback model where the longitudinal stress created by smooth muscle is transduced into tension-dependent YAP activity and growth of the intestine. However, the authors do not directly show that YAP is regulated by tension in this system, and the data do not exclude alternative models where HH activates YAP independent of tension.

Major comments

1. The claim that YAP localization and activity in muscle cells is regulated by a mechanical, rather than a biochemical, signal generated by ciliary HH signaling is not strongly supported. The perturbations the authors use to disrupt smooth muscle-generated forces (reducing ciliary signaling, HH activity, or ablating smooth muscle cells) also disrupt biochemical signals generated by the smooth muscle. The authors should either provide direct evidence that applying a force can bypass the requirement for HH (but not YAP) in this system, or they should clarify throughout the text and in the abstract that their data are also consistent with a biochemical connection between HH and YAP.

2. The spatial and functional relationships between the peri-epithelial mesenchymal cells that have activated nuclear YAP, the smooth muscle cells that respond to HH, and the epithelial cells of the intestine should be better explained, perhaps in a figure. Which of these cell types express Dermo1 Cre? Which have nuclear YAP? It would strengthen the paper to more clearly explain which results relate to each population, and this would clarify whether HH regulates YAP through a paracrine or autocrine mechanism.

3. The smooth muscle phenotype should be quantified in all genotypes, using a consistent metric for

smooth muscle organization, and any differences between genotypes should be addressed. For example, it is difficult to evaluate the authors' claim that smooth muscle organization is normal in YAP^{-/-}; TAZ^{+/-} embryos based on the images shown.

4. The YAP data should be quantified in the same way in all figures to facilitate comparison between different cell populations and genotypes.

5. Does cell death occur in the culture experiments after longitudinal cutting in Figure 3G? The overall morphology of the tissues in Figure 3G and Figure S10C is quite disrupted. Analysis of markers of apoptosis or necrosis would aid in the interpretation of these results.

6. Are the authors proposing that YAP causes intestinal elongation by both activating and orienting proliferation in the epithelium? Evidence that cell divisions are oriented would support this model. At minimum, the authors should propose mechanisms that could elongate the intestine downstream of YAP.

7. At the top of page 5: "We quantified the prevalence of apoptotic cells...(Figure S5)." No quantification is presented in Figure S5. Later on page 5, it is not clear how the measurements mentioned in the text ("increased variance of tangential alignment and increased radial distribution") relate to Figure 2B, which shows intensity. On page 7, Figure 6F should be Figure 3F.

Reviewer #2:

Remarks to the Author:

Overall the manuscript is well written and addresses a very important topic in the field of GI development (and disease).

Noteworthy results:

In agreement with previously published data, loss of Hh signaling results in defects of muscle development, short intestines and altered YAP localization. While the phenotypes are slightly different, mutations affecting primary cilia (required for Hh signaling) also result in altered smooth muscle development, YAP nuclear localization and shortened intestines. The finding that a YAP inhibitor can restore intestinal lengthening is of great interest in understanding the process of intestinal lengthening and how that may be leveraged towards driving intestinal lengthening in clinical settings.

The main concerns with the manuscript are:

- 1) The number of animals assessed in most of the data presented are very few (3-4 in most cases) and likely represent a single litter of animals.
- 2) Many different stages are shown throughout the figures, but there does not seem to be consistency or an explanation as to why a particular stage is shown and not others. Please see specific comments throughout for further explanation.
- 3) There are so many mutants included in the manuscript that it seems to this author dilute your story. Additionally, the inclusion of all these mutants without thorough examination of them is more distracting than helpful. This reviewer would recommend focusing on the key mutants that make your points and providing thorough studies of those rather than bits and pieces of many mutants.
- 4) This reviewer does not feel that you have ruled out contribution of muscle contraction as a force driving intestinal lengthening. Please see additional comments for further explanation.

Specific comments/concerns:

Page 2- Though smooth muscle instructs the formation of ridges, zigzags and villi in the chick intestine, it does not instruct the formation of epithelial villi in the mouse.

Show scale bars on all panels

Figure 1

Why is there a variation in stages shown? E14.5 in panel A, E16.5 in panel C, E13.5 in panel G. Please

include all stages or explain why a particular stage is shown and not the others.

What muscle layers are present in the esophagus at each stage? Does the esophagus show a similar growth pattern (not different at E13.5 and then by E14.5 there is a significant difference in length)? Is there a proliferation defect in the esophagus too? Why show only E16.5 for esophagus?

Did you look at proliferation at later stages or only at E13.5? Is the proliferation rate the same at all stages? It appears that only 3-4 samples were counted for each genotype—was this 3-4 different animals or different fields from the same animal? Is this really a sufficient number of samples to draw a conclusion? Did you examine proliferation in any of the other mutants- is the proliferation defect consistent in all cilia mutants?

If the proliferation is so drastically reduced, why is the circumference not affected?

Figure S1

How many litters were examined? Seem to be very few N...

Again, different stages are shown- E15.5 in B, but E18.5 in D and E16.5 and E17.5 in C. How did you determine the site to measure for proximal and distal? Diameters can be very different based on where you measure and with the shortened intestine, you would need to approximate a similar point as compared to the controls (i.e. 1/3 and 2/3 of total intestinal length).

Figure S2

B- only +/- and -/- embryos are shown. Why not +/+? The stage is noted at E13.5, but the embryos shown are E11.5-12 at most based on Theiler staging.

C The Tctn-/- do appear to have less Arl13b staining at the lower magnification. Have the number of primary cilia been quantified? Is there less Arl13b in the cilia (normal numbers of cilia) or are there fewer cilia? From these single panels, it also appears that the cilia that are present may be longer? If so, could longer cilia with more diffuse Arl13b just be harder to visualize than the shorter cilia with compact Arl13b staining? An overlay of Arl13b, acetylated tubulin and gamma-tubulin would be helpful.

D-F. It appears for some genotypes only 3 mice were examined.

F. Why is intestinal length for the Tctn3 mice only shown at E15.5?

Figure S4

Please include scale bars- the lower panels appear to be at a different magnification or are from different regions of intestine.

Again, why are different stages used? Esophagus is now shown at E15.5 (previous figure, it was shown at E16.5). Ift88 mutants are shown only at E14.5 and the QPCR data is from E15.5. Do Gli1 and Ptch1 go down in the IFT88 mutants and the other cilia mutants? Do they decrease in the Dermo1Cre; Smo f/f mutants?

Figure S5

Show scale bars. Control vs mutant intestines appear to be very different sizes- different stages or different regions? It is also difficult to see the cleaved caspase-3 stain, including a magnified inset might help.

p.5 typo: we detected no increased in apoptosis

Was cleaved caspase-3 assessed in any other mutants?

Figure 2

Panel A- The aSMA staining in the Cilk1-/- mutants looks like aSMA staining at E12.5. Is it possible that the Cilk1-/- are just delayed in development? What does the muscle staining look like at later stages?

Figure S6A- delayed development? What do later stages look like?

Figure S6B- mutant panels are a poor plane of section where the right side is twisted and glancing. These panels need to be replaced with a properly oriented section in order to make a comparison. The left side of the section that is not twisted/glancing looks more normal.

Figure 2D- Is the mutant just a little delayed? Showing a later stage might clear up this question...
Figure S7- The muscle layer shown here is called inner circular muscle or ICM. How did you ensure that regions shown are from the same area of intestine? Was your N of 3 from a single litter? Did you examine later stages? Does the outer longitudinal layer develop? If so, is it thinner?
This Smo phenotype is different from the cilia mutants. Why is there a difference (perhaps mentioned in one sentence of the discussion as differential processing of Gli- elaboration on this point would be helpful)? Do the cilia mutants each allow some degree of Hh signaling while the Smo mutants are complete loss of Hh signaling? Could you combine conditional cilia mutants for complete loss of Hh signaling in mesenchyme (Dermo1-Cre) or muscle (Myh11-Cre) specifically?
If Hh signaling is required for smooth muscle development, why does the ICM develop? Could the timing of Dermo1-Cre expression relative to Hh signaling in muscle explain why ICM develops or perhaps Dermo1-Cre does not recombine all muscle cells?
Did you try Myh11-Cre to recombine Smof/f and Cilk1f/f specifically in muscle vs. general mesenchyme?

Figure 3. Atomic force microscopy would be a more convincing measure of structural differences. Measuring opening angles on such tiny pieces of intestine calls for a lot of human interpretation which can be made less biased by measuring MANY blinded samples. 3-4 samples is not convincing. Did you examine cell death in the intestines? Was death limited to muscle cells in the DTA induced intestines or was there a general increase in cell death throughout the intestine? It is possible that an overall increase in cell death (or decreased proliferation) could account for the shortened length... Assuming increased numbers of measurements yields reliable differences in opening angles, could you correlate the opening angles to the intestinal length of the various mutants and controls? In other words, would larger opening angle yield a full-length intestine, an intermediate opening angle yield an intermediate length intestine and a small opening angle yield a short intestine?

Figure S9. This reviewer does not know enough about the ins and outs of staining with phosphorylated muscle myosin light chain kinase, but fetal intestines at E13.5 do in fact contract. In fact, contractions increase by E14.5 (the stage when you begin to note differences in lengthening). A great experiment would be to place your various mutants in culture and measure/live image their contractions.
p.7 typo: disruption of the smooth muscle by genetically ablation

Figure S10C- Muscle contractions are occurring in the uncut intestines and likely contribute to elongation. One other caveat is that opening of the intestine can be quite damaging—any place where the intestine was touched will not develop normally, therefore the act of cutting open the intestine may have prevented lengthening.

Figure 4. Please include an inset with magnifications of the YAP staining for panels in D. Does nuclear YAP decrease in cut open WT intestines? Please explain the significance of decreases in Edn1, Ano5, and Ccl11. What do changes in these markers represent?

Figure S11. Quantification would be helpful. It is difficult to see/assess nuclear YAP in these magnified panels.

Figure 5. It is interesting that mesenchymal loss of Yap/Taz results in decreased epithelial EdU (similar to mesenchymal loss of Cilk1). Perhaps this suggests that a loss of mesenchymal proliferation feeds into decreased proliferation of the epithelium as well— a mechanism to ensure coordinated growth rates of the two layers?

P.9 Typo: We propose that, at [as] in other tissues. The different effects...

Figure 6. What does the muscle look like in the Cilk mutants after YAP inhibition. Previous studies showed that YAP needed to be downregulated for muscle development. Is muscle development

rescued with YAP inhibition?

Reviewer #3:

Remarks to the Author:

Please see attached file with comments.

In their study, Ying Yang and co-authors examine the biomolecular and mechanobiological cues giving rise to intestinal and oesophageal elongation in the fetal mouse embryo (E13.5-E18.5). Using an impressive array of mutants, they find that cilia located on mesenchymal cells of the intestine are required for gut elongation, with the mutant being down by 40-60% in length compared to controls. Cilia activity is found to be necessary along with hedgehog (HH) to pattern a well-defined, circumferentially aligned smooth muscle ring. Malformation of this smooth muscle ring via a cilia, a HH-pathway defect, or by localized expression of diphtheria toxin results in growth defects. The measurement of opening angles of gut segments shows that the SMA ring contributes to residual circumferential stress, and that this contribution is reduced in all genetic manipulations that resulted in a malformed SMA ring. Yap levels are found to be decreased in the mutants (Cilk1 $-/-$, Dermo1CreSmo lox/lox), and conditional YAP KO in the mesenchyme results in shorter, less proliferating guts than controls. Boosting YAP signaling with XMU-MP-1 in Cilk1 $-/-$ intestines rescued elongation in culture. The authors conclude on the following sequence of events: the epithelium produces HH, which is transduced in the mesenchyme by the cilia to give rise to the circumferential smooth muscle ring, which produces force, activating YAP, and activating proliferation and gut lengthening. The authors also demonstrate that embryonic mouse intestines can be grown ex-vivo in matrigel (almost 2-fold elongation after 2-day culture).

I congratulate the authors on this very clear, rigorous, technically sound, exhaustive biomolecular investigation of gut growth in the fetal mouse. The results reinforce several recent studies showing that smooth muscle generated forces are key to gut and lung morphogenesis. I have the following comments:

- my main concern pertains to the fact that the authors do not explain why the force-YAP-proliferation pathway outlined should lead to an *elongation* of the organ, rather than to its thickening (diameter increase). Khalipina et al. ("Smooth muscle contractility causes the gut to grow anisotropically" J. Royal Soc. Interface 2019) have outlined how the Poisson effect (due to the incompressibility of the gut interior) transforms centripetally directed circumferential smooth muscle pressure (tonic or phasic) into elongation of the embryonic gut. Alternatively, the passive (non-muscle) anisotropic biomechanical characteristics of the gut can favor cell proliferation along the longitudinal axis; these anisotropic properties have also been characterized in Khalipina et al. for the embryonic chicken gut. Both of these mechanisms should be discussed because they complement the author's results.
- the embryonic mouse gut undergoes extensive elongation before the smooth muscle differentiates: at E12.5, before any signs of α -SMA can be uncovered, it is already an anisotropic 1 cm long – ~ 200 μ m diameter tube. It thus appears that the force-YAP-proliferation pathway is only the later – fetal - part of the story behind intestinal elongation. The authors should point this out and discuss what mechanisms might underlie early embryonic elongation, prior to SMA differentiation. Are other forces present at early stages that trigger YAP nuclear translocation, e.g., vitelline duct tension? Or could the YAP switch be "off" in early gut morphogenesis?
- the authors should discuss whether cilia- of HH-dependent mutations have been documented in short bowel syndrome.
- p.3 the authors should add representative pictures at a given age of B9d1, Inpp5e and Tctn3 mutant guts (whole mount) in Fig.S2 (perhaps in the blank space next to S2B), from which the morphological data of S2D-F was retrieved.
- it is interesting that guts could be grown in matrigel, while one would think that the elastic gel would act as a mechanical barrier to elongation. Have the authors tried growth in plain liquid BGJb medium, can they comment on the necessity for a gel? The authors should also add the reference to the first

paper that reported embryonic mouse gut growth using this medium, Duh. et al. “EGF regulates early embryonic mouse gut development in chemically defined organ culture”, *Pediatr. Res.* 2000.

- p.6 “Smooth muscle was positive for phospho-MLC at E18.5, but negative at E13.5 (Figure S9), suggesting that the circumferential smooth muscle promotes intestinal elongation before it becomes contractile.” is not correct, because propagating contractile waves can be observed propagating along the whole midgut at E13.5 (but not at E12.5) – the authors can verify this by time-lapse imaging E13.5 guts or by checking this reference: R. Roberts et al. “The first intestinal motility patterns in fetal mice are not mediated by neurons or interstitial cells of Cajal”, *J. Phys.* 2010. The circular smooth muscle has tonic (time-independent) and phasic (dynamic) contractile components that are both active in generating force on the gut interior (see Khalipina et al.), and thus in activating YAP – more on that below.

- The opening angles (Fig.3, p.7) are measured at room temperature in DPBS without calcium, a situation where the smooth muscle is not able to contract: it is thus passive (non-muscle) stress due to the shear presence of this layer of tensed aligned cells & ECM that is measured. The authors should clarify that this is however only a minor component of the stress, the other – main one – being the active tonic and phasic force generated when SMA is active (as in a developing mouse embryo at 37°C). Although these have not been measured by the authors, the result would very probably be that the various mutants generate less active stress. This should be pointed out and the authors should replace the term “residual stress” in the remainder of the manuscript by “force” (example p.7: “raised the interesting possibility that force produced by the circumferential smooth muscle promotes longitudinal growth”).

- the unaltered smooth muscle layer of the E13.5 *Yap lox/lox* could usefully be moved from S12B to Fig.5 (a neat blank space is left).

- p.9 please clarify what is “GLI”

- the title of the manuscript is inaccurate: experiments have been performed on the small intestine and the esophagus, i.e., on the gastrointestinal tract. Other smooth muscle lined tubular organs are of course the blood vessels, and although an extension of the mechanism outlined sounds exciting, it remains speculative. Perhaps a title more reflective of the biochemical effort that went in the study on the HH-cilia-smooth muscle-force-YAP pathway would be more informative.

Dr. Nicolas R. Chevalier

Dear Reviewers,

I and my co-authors thank you for your helpful insights. We deeply appreciate your comments as they led to extensive additions that substantively strengthened and extended the scope of the manuscript. We were encouraged to learn that you found that, “This study... provides valuable data on how tubes are organized and elongate,” (Reviewer 1), and that “Overall the manuscript is well written and addresses a very important topic in the field of GI development (and disease),” (Reviewer 2) and that Dr. Chevalier wrote, “I congratulate the authors on this very clear, rigorous, technically sound, exhaustive biomolecular investigation of gut growth in the fetal mouse,” (Reviewer 3). We have spent the last few months generating data and making revisions with the goal of further improving the quality of the work and addressing your concerns. We were asked to focus on four important points, described immediately below. We include descriptions farther below responding to each of your other points. Critiques are delineated in blue font.

1. Please include further work, if possible, to show the effect of direct force affecting Hh signalling not biochemical/signal activation.

To modulate force, we made use of the in vitro intestinal elongation system that we developed for this manuscript. By changing the concentration of growth factor-reduced Matrigel from 50-90%, we can generate matrix with varied stiffness (i.e., elastic modulus). Previous work from others has indicated that these changes in matrix composition result in an approximately four-fold change in stiffness¹⁻⁵. We found that increasing matrix stiffness in vitro increased nuclear levels of YAP, and was sufficient to partially restore longitudinal growth of *Cilk1*^{-/-} intestine. These new data are included as the new Figure 6C-E.

2. Please also address points regarding elongation vs. diameter enlarging but we would note here that full mechanism is not expected, just clarity.

Although we appreciated the opportunity to address the question about how altered ciliary Hedgehog signaling affects longitudinal growth but not radial growth of the mouse intestine with a textual revision, we attempted to address this question with new experimental data as well. More specifically, we immunofluorescently stained E13.5 intestinal sections for phospho-histone H3 (pHH3), α -tubulin and γ -tubulin. Consistent with our previous results, these experiments confirmed that proliferation is reduced in *Cilk1*^{-/-} intestines. As pHH3 staining indicates mitotic cells, α -tubulin stains the mitotic spindle, and γ -tubulin

stains the spindle poles, we were able to discern the division plane of mitotic cells within the developing intestine. Interestingly, we found that more cells divide longitudinally than radially in wild-type intestine. In *Cilk1* mutant intestine, there was no significant reduction in radially dividing cells, but there was a reduction in longitudinally dividing cells. These experiments suggest a cell biological mechanism by which *CILK1* may contribute to longitudinal, but not radial, growth of the intestine. We are motivated by these findings to investigate why radial divisions do not depend on *CILK1* function in future experiments. These novel findings are included in the revised Figure S1E-F and described in the associated text in the Results section.

3. Please also provide more details on conclusions that can be drawn from mutants and add more quantitation and similar readouts to enable a comparison between phenotypes (for example, consistency of YAP measurements).

In the revised manuscript, we applied a consistent metric to quantify smooth muscle organization (included in the revised Figures 2E, S6B, S6D, S6F, and S12C), and nuclear YAP (in revised Figures 6D, S11A, S11C and S11E), thereby facilitating comparisons between different phenotypes and genotypes.

4. Please also provide 'n' values for all experiments.

We apologize for this omission. In the revised manuscript, the 'n' for all experiments is noted in the figure legends.

REVIEWER COMMENTS

Reviewer #1:

Yang et al. provide evidence for a novel mechanism in which ciliary signaling elongates the mouse intestine and esophagus by activating YAP in circumferential smooth muscle cells. They show that ciliary and SMO mutants, which disrupt intestinal elongation, impact smooth muscle cell organization and proliferation, and that the primary requirement for cilia and SMO is in the mesenchymal cells surrounding the epithelial tubes. Defects in intestinal elongation are recapitulated by ablation of smooth muscle cells, further indicating that the smooth muscle is required for intestinal elongation. By making longitudinal cuts in the intestine, the authors show that the smooth muscle is a source of residual stress in the intestine at the time of elongation, and that this residual stress requires cilia and HH signaling. Using an in vitro culture system, the authors show that longitudinal cuts that reduce residual stress also prevent intestinal elongation, although these cuts appear to cause cell damage that could complicate the interpretation of these results.

The authors go on to show that ciliary mutants have reduced nuclear localization of YAP in peri-epithelial stromal cells and reduced expression of several YAP target genes. Intestines homozygous for YAP and heterozygous for TAZ in the mesenchyme have reduced elongation and decreased cell proliferation, indicating that YAP activity in the mesenchyme is required for elongation and that YAP is an important target of HH activity. They show using their cell culture system that activating YAP rescues intestinal elongation in a *Cilk* mutant intestine, indicating that reduced YAP activity is the cause of the intestinal elongation defects in cilia mutants.

This study describes several new genetic tools and provides valuable data on how tubes are organized and elongate. The results support the model that ciliary signaling elongates

the intestine by directing cellular organization and YAP signaling in the circumferential smooth muscle layer. While the role of HH in intestinal elongation was known previously, the authors now link this signal to altered changes in YAP activity and show that YAP regulates intestinal elongation downstream of HH, linking two major signaling pathways to changes in tissue shape and mechanics. Because reduced intestinal elongation is associated with reduced proliferation and disrupted smooth muscle organization, the authors posit a mechanical feedback model where the longitudinal stress created by smooth muscle is transduced into tension-dependent YAP activity and growth of the intestine. However, the authors do not directly show that YAP is regulated by tension in this system, and the data do not exclude alternative models where HH activates YAP independent of tension.

Major comments

1. The claim that YAP localization and activity in muscle cells is regulated by a mechanical, rather than a biochemical, signal generated by ciliary HH signaling is not strongly supported. The perturbations the authors use to disrupt smooth muscle-generated forces (reducing ciliary signaling, HH activity, or ablating smooth muscle cells) also disrupt biochemical signals generated by the smooth muscle. The authors should either provide direct evidence that applying a force can bypass the requirement for HH (but not YAP) in this system, or they should clarify throughout the text and in the abstract that their data are also consistent with a biochemical connection between HH and YAP.

As suggested by the reviewer, we have clarified in the text that other models, such as a biochemical connection between HH and YAP signaling, are not ruled out by our findings. In addition, as described in response to the editor's point 1, we include new experiments that indicate that increasing the matrix elastic modulus *in vitro* is sufficient to increase nuclear levels of YAP and partially bypass the requirement for HH signaling in the elongation in *Cilk1*^{-/-} intestines.

2. The spatial and functional relationships between the peri-epithelial mesenchymal cells that have activated nuclear YAP, the smooth muscle cells that respond to HH, and the epithelial cells of the intestine should be better explained, perhaps in a figure. Which of these cell types express Dermo1 Cre? Which have nuclear YAP? It would strengthen the paper to more clearly explain which results relate to each population, and this would clarify whether HH regulates YAP through a paracrine or autocrine mechanism.

Dermo1-Cre marks mesenchymal cells, including the smooth muscle cells (as revealed by reporter activation included in Figure S4A). Nuclear YAP accumulates in mesenchymal cells internal to the smooth muscle (indicated in, among others, Figure 4A). Previous work has demonstrated that mesenchymal cells, especially those proximal to the intestinal epithelium, respond to HH signals produced by the epithelium^{6,7}. As smooth muscle patterning depends on HH signaling to the mesenchyme and YAP activity in cells interior to the smooth muscle depend on smooth muscle integrity, we propose that HH and YAP signaling are active in some of the same mesenchymal cells, but that HH regulates YAP via an indirect mechanism depending on smooth muscle patterning. As suggested by the reviewer, we have explained these relationships in the text of the manuscript more thoroughly and added a figure to further delineate the populations (the new Figure S11A).

3. The smooth muscle phenotype should be quantified in all genotypes, using a consistent metric for smooth muscle organization, and any differences between genotypes should be

addressed. For example, it is difficult to evaluate the authors' claim that smooth muscle organization is normal in YAP^{-/-}; TAZ^{+/-} embryos based on the images shown.

As suggested (and as detailed above in the responses to the editor's points), we applied a consistent metric to quantify smooth muscle organization (as revealed by α SMA staining). These metrics, included in the revised Figures 2E, S6B, S6D, S6F, and S12C, allow the reader to compare smooth muscle organization across genotypes.

4. The YAP data should be quantified in the same way in all figures to facilitate comparison between different cell populations and genotypes.

Also as detailed above in the responses to the editor's points, we have quantified nuclear YAP in a consistent way throughout the revised Figures 6D, S11A, S11C and S11E to facilitate comparisons across different samples.

5. Does cell death occur in the culture experiments after longitudinal cutting in Figure 3G? The overall morphology of the tissues in Figure 3G and Figure S10C is quite disrupted. Analysis of markers of apoptosis or necrosis would aid in the interpretation of these results.

We concur that the longitudinal incision of intestines more dramatically disrupts elongation and, perhaps consequently, overall morphology. For the revised manuscript, we immunostained cut and uncut intestines for cleaved Caspase-3, a marker of apoptosis. Interestingly, we did not observe any no change in apoptosis caused by longitudinal incision. These data are included in the new Figure S9D.

6. Are the authors proposing that YAP causes intestinal elongation by both activating and orienting proliferation in the epithelium? Evidence that cell divisions are oriented would support this model. At minimum, the authors should propose mechanisms that could elongate the intestine downstream of YAP.

As the intestine elongates more than it radially expands, we assume that proliferation or cell migration within the developing intestine are organized to drive growth predominately along the longitudinal axis. We describe some textual revisions and additional data investigating whether CILK1 controls the axis of cell division above in the comments in response to the editor's points. While the data included in Figure 5C-E indicate that YAP activation promotes intestinal proliferation, we do not currently have any data that YAP activity affects the orientation of cell division. As suggested by the reviewer, we now include discussion of whether YAP might direct cell division or cell migration to promote longitudinal growth in the intestine, either directly or indirectly.

7. At the top of page 5: "We quantified the prevalence of apoptotic cells...(Figure S5)." No quantification is presented in Figure S5. Later on page 5, it is not clear how the measurements mentioned in the text ("increased variance of tangential alignment and increased radial distribution") relate to Figure 2B, which shows intensity. On page 7, Figure 6F should be Figure 3F.

Thank you for pointing out these errors. We have fixed them. More specifically, in the revised manuscript, we have added the quantification of apoptotic cells (included in the new Figure S5A). The measurement of variance of tangential alignment of *Cilk1*^{-/-} intestines are now

included in Figure 2A. The radial distribution measurements are reflected in the intensity measurements included in Figure 2B. We have changed "Figure 6F" to "Figure 3F" on page 7.

Reviewer #2:

Overall the manuscript is well written and addresses a very important topic in the field of GI development (and disease).

Noteworthy results: In agreement with previously published data, loss of Hh signaling results in defects of muscle development, short intestines and altered YAP localization. While the phenotypes are slightly different, mutations affecting primary cilia (required for Hh signaling) also result in altered smooth muscle development, YAP nuclear localization and shortened intestines. The finding that a YAP inhibitor can restore intestinal lengthening is of great interest in understanding the process of intestinal lengthening and how that may be leveraged towards driving intestinal lengthening in clinical settings.

The main concerns with the manuscript are:

1) The number of animals assessed in most of the data presented are very few (3-4 in most cases) and likely represent a single litter of animals.

All experiments represent multiple litters (as, depending on the genetics, informative genotypes are 12.5% or 25% on average of C57BL/6J litters, which is typically 6-7 embryos).

2) Many different stages are shown throughout the figures, but there does not seem to be consistency or an explanation as to why a particular stage is shown and not others. Please see specific comments throughout for further explanation.

We have added explanations as to why particular stages are shown in the text of the revised manuscript and below.

3) There are so many mutants included in the manuscript that it seems to this author dilute your story. Additionally, the inclusion of all these mutants without thorough examination of them is more distracting than helpful. This reviewer would recommend focusing on the key mutants that make your points and providing thorough studies of those rather than bits and pieces of many mutants.

Multiple ciliary mutants in Figure S2 were used to demonstrate that the short intestine is shared by several mutants that have defective cilia in common, suggesting that the intestinal elongation phenotype is a result of defective ciliary function. Most of our study focuses on *Cilk1* knockout embryos. Conditional *Cilk1* mutations mediated by *Shh*^{Cre} and *Dermo1*^{Cre} are used to dissect the cell type specific functions of CILK1 in regulating the intestinal length and the results indicated that the mesenchymal CILK1 drives the intestinal elongation. *Dermo1*^{Cre} mediated depletion of *Smo* and *Yap* are used to demonstrate the requirement of mesenchymal SMO and YAP for smooth muscle patterning, residual stress, and the elongation of the developing intestine as potential downstream mediators of ciliary functions. Genetic ablation of smooth muscle cells using *Myh11*^{Cre-EGFP} *R26*^{DTA176/+} is used to demonstrate the contribution of smooth muscle cells to intestinal residual stress and elongation. To improve the clarity of the manuscript, we have revised the text and further explained how the different genetic models make complementary biological points.

4) This reviewer does not feel that you have ruled out contribution of muscle contraction as a force driving intestinal lengthening. Please see additional comments for further explanation.

We concur. As we have not ruled out contribution of muscle contraction to intestinal elongation, we have added comment to the Results and Discussion sections acknowledging that the experiments described in this work do not indicate that muscle contraction does not contribute to the forces generated within the intestine or to its longitudinal growth.

Specific comments/concerns: Page 2- Though smooth muscle instructs the formation of ridges, zigzags and villi in the chick intestine, it does not instruct the formation of epithelial villi in the mouse.

We concur. To avoid misleading the reader, we have removed the reference to villi, as our work does not investigate villus development.

Show scale bars on all panels

We have included scale bars in the bottom image of each panel and in insets. We have updated the methods section and figure legends to report on scale bars.

Figure 1. Why is there a variation in stages shown? E14.5 in panel A, E16.5 in panel C, E13.5 in panel G. Please include all stages or explain why a particular stage is shown and not the others. What muscle layers are present in the esophagus at each stage? Does the esophagus show a similar growth pattern (not different at E13.5 and then by E14.5 there is a significant difference in length)? Is there a proliferation defect in the esophagus too? Why show only E16.5 for esophagus?

In the revised manuscript, we include measurement of esophageal length at E14.5 that indicate that differences in the length in *Cilk1*^{-/-} esophagi emerge by E16.5 (Figure 1D). To investigate the contributors to the altered length at E16.5, we investigated the patterning of esophagi at a timepoint earlier than E16.5. To be more specific, we assessed smooth muscle patterning (Figure 2D-E) and proliferation (Figure S1G) at E14.5 in esophagi. These results are consistent with the model that the smooth muscle patterned in a CILK1-dependent way promotes cellular proliferation to drive the elongation of tubular organs including both the intestine and esophagus, although the precise timing of that difference in proliferation may differ in different organs. As posited by the reviewer, smooth muscle patterning is not clearly altered in E13.5 *Cilk1*^{-/-} esophagi, and thus we included data about the effects of CILK1 on smooth muscle patterning at E14.5 for esophagi.

Did you look at proliferation at later stages or only at E13.5? Is the proliferation rate the same at all stages? It appears that only 3-4 samples were counted for each genotype—was this 3-4 different animals or different fields from the same animal? Is this really a sufficient number of samples to draw a conclusion? Did you examine proliferation in any of the other mutants- is the proliferation defect consistent in all cilia mutants?

We focused on assessing proliferation at E13.5 in the intestine and at E14.5 in the esophagus as these are stages just prior to the emergence of altered length in these organs

in the Cilk mutants. For each genotype, 3-4 animals were used. In Figure 1H and Figure S1G, each dot represents data from one animal. For each animal, EdU-positive and total cell numbers were quantified from three whole cross sections and averaged, as recommended by a biostatistician consultant. As variance is low, statistical tests demonstrate significance with the number of samples analyzed. We did observe that EdU incorporation were also reduced in E14.5 *Cilk1*^{-/-} esophagus. As suggested by the reviewer, we focused on analyzing *Cilk1*^{-/-} and *Dermo1*^{Cre} *Cilk1*^{-/-} embryos. The EdU results were shown from these two mutants in the revised version as the new Figure 1G,H and S1G. In addition, we have complemented EdU incorporation data with assessing mitotic cells using phospho-histone H3 staining.

If the proliferation is so drastically reduced, why is the circumference not affected?

It remains an intriguing observation that ciliary HH signaling appears to be critical for longitudinal, but not circumferential, growth. In the revised version and as described above, we assessed the division plane of mitotic cells in E13.5 intestine by co-staining for phospho-histone H3, α -tubulin and γ -tubulin. These new data are included in the revised Figure S1E-F. Consistent with the previously noted reduction in proliferation in *Cilk1*^{-/-} intestines, the total mitotic cell number was reduced. As might be expected for a rapidly elongating tubular tissue, more cells were dividing longitudinally than radially in the intestine. Notably, radial divisions were not reduced in *Cilk1*^{-/-} intestines. These results suggest that CLK1 is preferentially important for the longitudinally-oriented cell division.

Figure S1 How many litters were examined? Seem to be very few N... Again, different stages are shown- E15.5 in B, but E18.5 in D and E16.5 and E17.5 in C.

Each dot represents one animal. The phenotypes in Figure S1B were seen in all the litters (>20) that have been assessed. Data presented in Figure S1C are from 7 litters (41 embryos in total). Data presented in Figure S1D are from 2 litters (17 embryos in total). The multiple stages assessed suggest that the negative data included in in Figure S1B-D are durable.

How did you determine the site to measure for proximal and distal? Diameters can be very different based on where you measure and with the shortened intestine, you would need to approximate a similar point as compared to the controls (i.e. 1/3 and 2/3 of total intestinal length).

We certainly concur that intestinal diameter varies depending on anterioposterior position. Therefore, we measured the diameters at two anatomical landmarks, the gastrointestinal boundary and the ileocecal junction. We have included this information in the revised Figure S1C legend.

Figure S2B- only +/- and -/- embryos are shown. Why not +/+? The stage is noted at E13.5, but the embryos shown are E11.5-12 at most based on Theiler staging.

Previously, we did not include *Tctn3*^{+/+} embryo photomicrographs because they were indistinguishable from those of *Tctn3*^{+/-} embryos. In the revised manuscript, we have included a photomicrograph of a *Tctn3*^{+/+} littermate to generate the new Figure S2B. We confirmed that the embryos depicted were at E13.5. As the reviewer no doubt realizes, stage can vary depending on genetic background, and these embryos are C57BL/6J.

C The *Tctn*^{-/-} do appear to have less Arl13b staining at the lower magnification. Have the number of primary cilia been quantified? Is there less Arl13b in the cilia (normal numbers of cilia) or are there fewer cilia? From these single panels, it also appears that the cilia that are present may be longer? If so, could longer cilia with more diffuse Arl13b just be harder to visualize than the shorter cilia with compact Arl13b staining? An overlay of Arl13b, acetylated tubulin and gamma-tubulin would be helpful.

Consistent with a previous report showing that TCTN3 is required for ciliogenesis in other tissues and MEFs⁸, we found a reduction of ciliated cells in E13.5 intestines from 93% in *Tctn3*^{+/-} intestines to 67% in *Tctn3*^{-/-} intestines, as assessed using co-staining of acetylated tubulin and γ -tubulin. In the ciliated cells, 100% cilia were positive for ARIL13B in *Tctn3*^{+/-} and 50% in *Tctn3*^{-/-}. Therefore, 93% cells have ARIL13B-positive cilia in *Tctn3*^{+/-} and 33.5% in *Tctn3*^{-/-}. To focus the analysis of *Tctn3*^{-/-} ciliary phenotypes on intestinal development and because these effects are consistent with prior data from our lab on other transition zone mutants and others on a different *Tctn3* mutant, we did not comment extensively on these data in the manuscript. As perceptively noted by the reviewer, the ARL13B-positive cilia in *Tctn3*^{-/-} embryos are longer on average than those in *Tctn3*^{+/-} embryos. Ciliary length in *Tctn3*^{-/-} embryos also shows greater variance than in *Tctn3*^{+/-} embryos. These ciliary length effects are similar to those described previously for other transition zone mutants. In the revised manuscript, we added the overlaid Arl13b, acetylated tubulin and gamma-tubulin staining, as the reviewer suggested, to generate the new Figure S2C.

D-F. It appears for some genotypes only 3 mice were examined.

Each dot represents one animal. In Figure S1D-F, for each genotype at each embryonic age, the intestinal length was measured from more than 3 embryos except for *Inpp5e*^{+/+} and *Inpp5e*^{-/-} at E16.5. Given the small amount of variance in the phenotypes, the statistical tests indicate that the differences, given the number of samples measured, are significant.

F. Why is intestinal length for the *Tctn3* mice only shown at E15.5?

We conceive of analysis of the intestinal elongation phenotype of ciliary mutants apart from *Cilk1* mutants as a means of assessing whether this phenotype is unique to *Cilk1* mutants or shared by other ciliary mutants. Thus, we used *Tctn3* mutants to assess whether they, like *Cilk1* mutants, exhibit a shortened intestine. We did assess intestinal elongation phenotypes at other stages in *Tctn3* mutants, and included this stage as this is the timepoint at which we have the largest number of samples to most firmly make the point described above.

Figure S4 Please include scale bars- the lower panels appear to be at a different magnification or are from different regions of intestine. Again, why are different stages used? Esophagus is now shown at E15.5 (previous figure, it was shown at E16.5). *Ift88* mutants are shown only at E14.5 and the QPCR data is from E15.5. Do *Gli1* and *Ptch1* go down in the *IFT88* mutants and the other cilia mutants? Do they decrease in the *Dermo1Cre*; *Smo* *f/f* mutants?

Thank you for noticing this omission. The scale bars have not been included. The lower panels are slightly oblique, accounting for their larger size on one axis. Most of the analysis

of *Dermo1^{Cre} Cilk1^{lox/lox}* embryos was mostly conducted at E15.5, and thus we conducted the the esophagus immunofluorescence analysis and qRT-PCR at this stage. As alluded to by the reviewer qRT-PCR data, included here as Figure 1 for reviewers, reveals that mRNA levels of *Gli1* and *Ptch1* were reduced in *Tctn3^{-/-}* intestines compared to controls (*Tctn3^{+/+}* and *Tctn3^{+/-}* intestines) and *Dermo1^{Cre} Smo^{lox/lox}* intestines compared to control (*Smo^{lox/lox}* intestines).

Figure 1 for reviewers. (Left) Relative levels of *Gli1* and *Ptch1* mRNA in E13.5 intestines of *Tctn3^{+/+}*, *Tctn3^{+/-}* and *Tctn3^{-/-}* embryos. (Right) Relative levels of *Gli1* and *Ptch1* mRNA in E13.5 intestines of *Smo^{lox/lox}* and *Dermo1^{Cre} Smo^{lox/lox}* intestines. * $p < 0.05$, ** $p < 0.01$ by two-way ANOVA Sidak's multiple comparisons test.

Figure S5 Show scale bars. Control vs mutant intestines appear to be very different sizes- different stages or different regions? It is also difficult to see the cleaved caspase-3 stain, including a magnified inset might help. p.5 typo: we detected no increased in apoptosis Was cleaved caspase-3 assessed in any other mutants?

Thank you for pointing out our oversight. We have added scale bars and magnified insets to the revised Figure S5 and increased gain to improve visibility. We have also fixed the typo in the text of the revised manuscript. Less than 0.4% of cells are apoptotic in the control intestine and we observed no increase in apoptosis in *Dermo1^{Cre} Cilk1^{lox/lox}* intestine (Figure S5B). These results suggest that the elongation driven by cilia is unlikely through inhibiting programmed cell death. We did not check cleaved caspase-3 (CC3) staining in mutants other than *Dermo1^{Cre} Cilk1^{lox/lox}* as apoptosis was not the focus of our manuscript.

Figure 2 Panel A- The aSMA staining in the *Cilk1^{-/-}* mutants looks like aSMA staining at E12.5. Is it possible that the *Cilk1^{-/-}* are just delayed in development? What does the muscle staining look like at later stages?

Because differences in intestinal length in *Cilk1^{-/-}* embryos is evident by E14.5, we focused on analyzing the smooth muscle development prior to that point. We did not assess the α SMA staining at later stages after elongation defects are evident as the causality is less clear once multiple defects are present.

Figure S6A- delayed development? What do later stages look like?

As *Tctn3* mutants were used only to establish whether other ciliary mutants affecting intestinal elongation compromised smooth muscle development, we focused on analyzing the smooth muscle at E13.5, at the onset of differences in intestinal length.

Figure S6B- mutant panels are a poor plane of section where the right side is twisted and glancing. These panels need to be replaced with a properly oriented section in order to make a comparison. The left side of the section that is not twisted/glancing looks more normal.

In the revised manuscript, we replaced Figure S6B with a properly oriented section. In addition, we added quantification of α SMA intensity distributions to Figure S6B, S6D and S6F.

Figure 2D- Is the mutant just a little delayed? Showing a later stage might clear up this question...

We have compared the E14.5 esophagi to those at E16.5 (Figure S6E) and E18.5 (Figure 2 for reviewers), and the phenotypic consequences of loss of CILK1 do not recover.

Figure 2 for reviewers. Immunofluorescence staining of E18.5 intestine of indicated genotypes for smooth muscle (α SMA, green) and nuclei (Hoechst, blue). Scale bars, 25 μ m.

Figure S7- The muscle layer shown here is called inner circular muscle or ICM. How did you ensure that regions shown are from the same area of intestine? Was your N of 3 from a single litter? Did you examine later stages? Does the outer longitudinal layer develop? If so, is it thinner? This Smo phenotype is different from the cilia mutants. Why is there a difference (perhaps mentioned in one sentence of the discussion as differential processing of Gli-elaboration on this point would be helpful)? Do the cilia mutants each allow some degree of Hh signaling while the Smo mutants are complete loss of Hh signaling? Could you combine conditional cilia mutants for complete loss of Hh signaling in mesenchyme (Dermo1-Cre) or muscle (Myh11-Cre) specifically? If Hh signaling is required for smooth muscle development, why does the ICM develop? Could the timing of Dermo1-Cre expression relative to Hh signaling in muscle explain why ICM develops or perhaps Dermo1-Cre does not recombine all muscle cells? Did you try Myh11-Cre to recombine Smo^{f/f} and *Cilk1*^{f/f} specifically in muscle vs. general mesenchyme?

We appreciate the extensive questions and ideas that the reviewer has regarding our work. To ensure approximately same segments of the intestines in different embryos were compared, we embedded control and experimental intestines in parallel and selected sections the same distance from the gastrointestinal border. We have expanded the number of embryos included in Figure S7B-C which are not from a single litter. We have focused on assessing smooth muscle development before any impacts on intestinal elongation. Therefore, we have restricted our analysis of smooth muscle development in *Dermo1^{Cre} Smo^{lox/lox}* embryos to E13.5, one day before the emergence of an intestinal length phenotype. It will be interesting to assess outer longitudinal smooth muscle development as well, but as our focus on the phenotype of *Dermo1^{Cre} Smo^{lox/lox}* embryos was on intestinal elongation, we focused on phenotypes that preceded the impacts on elongation. As the reviewer correctly notes, the phenotypic consequences of loss of SMO and those caused by ciliary disruption are distinct. We and others have investigated those differences previously in other developmental contexts. For example, during both neural tube and limb bud development, loss of ciliary proteins and loss of SMO also lead to distinct phenotypes^{9,10}. In these other developmental contexts, these differences are caused by loss of SMO causing the HH pathway to be turned off (loss of GLI activator and presence of GLI repressor) while disruption of cilia cause the loss of both GLI activator and GLI repressor (a condition that we conceptualize as the HH pathway being in neutral). The transcriptional consequences of the HH pathway being off and being in neutral are distinct, and these differences are thought to underlie the distinct phenotypic consequences. We do not detect residual HH signaling in many cilia loss of function phenotypes that is further mitigated by additional cilia-associated mutations (as we previously reported in¹¹). It is likely that, by extension, these differences in transcriptional output also underlie the distinct phenotypic consequences in intestinal development caused by loss of SMO and by disruption of cilia. We now include discussion of these possibilities on page 10 of the revised manuscript. We do not conclude that HH signaling is required for smooth muscle differentiation, but rather for the patterning of the gut mesenchyme to direct smooth muscle differentiation to specific domains. The formation of the ICM may indicate that a portion of the muscle is specified via means that are independent of ciliary HH signaling, or that, as proposed by the reviewer, *Dermo1^{Cre}* results in ablation of SMO after some ICM has been specified. Use of the lineage reporter, included in Figure S4, suggests that *Dermo1^{Cre}* can effect recombination throughout the mesoderm. However, it remains possible that other conditional alleles, potentially including *Smo^{lox/lox}* could be more recalcitrant to recombination. We did not use *Myh11^{Cre}* to recombine *Smo^{lox/lox}* or *Cilk1^{lox/lox}* as these experiments would address the functions of SMO and CILK1 at a later stage than was our focus in this work. Future experimentation will address the interesting question of how SMO and CILK1 function within the smooth muscle.

Figure 3. Atomic force microscopy would be a more convincing measure of structural differences. Measuring opening angles on such tiny pieces of intestine calls for a lot of human interpretation which can be made less biased by measuring MANY blinded samples. 3-4 samples is not convincing.

We attempted to measure intestinal stiffness using atomic force microscopy. Atomic force microscopy is well suited for measuring tissue stiffness but does not assess radial or centripetal forces within a tissue, as opposed to other strain, contractile forces or elastic moduli. To distinguish residual strain of radial forces within the gut wall, we measured

opening angle, an established method to measure residual stresses in tubular tissues such as blood vessels and the gut, as well as in non-biological tubular materials. In the data included in Figure 3D-F, 3-4 animals of each genotype were used. For each intestine, 5-15 segments of approximately 1 mm length were cut. A single radical cut was made through each segment. Immediately after the radical cut was made, a phase-contrast photomicrograph of the opened segment was taken. Using the Angle Tool in ImageJ, the opening angle was measured from the photomicrograph between two lines intersecting the midpoint of the inner epithelial surface and extended to the inner edges of the cut wall, as shown in Figure 3C. Thus, for each intestine, the opening angles from 5-15 segments were measured using ImageJ. The median angles of each intestine were used in plotting.

Did you examine cell death in the intestines? Was death limited to muscle cells in the DTA induced intestines or was there a general increase in cell death throughout the intestine? It is possible that an overall increase in cell death (or decreased proliferation) could account for the shortened length...

We agree with the reviewer that possible non-specific cell ablation using DTA is a general concern with this method. We counted the total cell number in the cross-sections and found no additional significant reduction of cell number in *Myh11*^{Cre-EGFP} *R26*^{DTA176/+} intestines. If the reviewer deems it important, we can include these data in the manuscript.

Assuming increased numbers of measurements yields reliable differences in opening angles, could you correlate the opening angles to the intestinal length of the various mutants and controls? In other words, would larger opening angle yield a full-length intestine, an intermediate opening angle yield an intermediate length intestine and a small opening angle yield a short intestine?

The opening angle experiment was performed at E13.5 when there was no significant length alterations in the intestines. There may be a correlation between the opening angle and the length across genotypes; larger reduction in the opening angle is associated with larger reduction in intestinal length. The average median angle of *Myh11*^{Cre-EGFP} *R26*^{DTA176/+} is 82% of *Myh11*^{Cre-EGFP} (94.1 vs 114.4) and the length is 80% (20.4 vs 25.5 in E15.5). The average median angle of *Cilk1*^{-/-} is 74% of *Cilk1*^{+/-} (87.4 vs 118.5) and the length is 52% (11.3 vs 21.8 in E14.5). The average median angle of *Dermo1*^{Cre} *Smo*^{lox/lox} is 68% of *Dermo1*^{Cre} *Smo*^{lox/WT} (83.4 vs 122.2) and the length is 45% (8.7 vs 19.4 in E14.5). Comparison across genotypes is complicated by multiple hypothesis testing.

Figure S9. This reviewer does not know enough about the ins and outs of staining with phosphorylated muscle myosin light chain kinase, but fetal intestines at E13.5 do in fact contract. In fact, contractions increase by E14.5 (the stage when you begin to note differences in lengthening). A great experiment would be to place your various mutants in culture and measure/live image their contractions.

Thank you for the correction. Indeed, we have also observed that both the frequency and amplitude of the intestinal smooth muscle contraction increases as development proceeds. Reflecting this correction, we have modified the text in page 7 in the revised manuscript to note that circular smooth muscle contraction can contribute to force generation in the developing intestine and its elongation. To explore the contribution of muscle contractions in driving the elongation at this development stage (E13.5), we newly tested the effects of

myosin inhibitor Y-27632 on cultured intestine elongation. Pharmacological inhibition of myosin function did not attenuate intestinal elongation, suggesting that myosin function is not critical for intestinal elongation. These new data are included in the new Figure S10A. We agree that it be interesting to measure contractions of the various mutants studied in this work. However, as this particular study is not focused on the role of contractions in intestinal elongation, we have saved those experiments for a future report.

p.7 typo: disruption of the smooth muscle by genetically ablation

Thank you for noting this typo. We have corrected it in the revised manuscript.

Figure S10C- Muscle contractions are occurring in the uncut intestines and likely contribute to elongation. One other caveat is that opening of the intestine can be quite damaging—any place where the intestine was touched will not develop normally, therefore the act of cutting open the intestine may have prevented lengthening.

We concur. We now include commentary on the roles of smooth muscle contraction in promoting intestinal elongation within the revised Discussion. In addition, we have included in the revised page 7 discussion of data proliferation rates and apoptosis levels in longitudinally incised intestines. These data are also included in the revised Figure S9C-D and Figure 3G-H.

Figure 4. Please include an inset with magnifications of the YAP staining for panels in D. Does nuclear YAP decrease in cut open WT intestines? Please explain the significance of decreases in *Edn1*, *Ano5*, and *Ccl11*. What do changes in these markers represent?

As we have included E14.5 esophagi in the revised manuscript Figure 4D, we concluded that magnified insets were not required. Please let us know if you would like us to include these. As suggested, we measured nuclear YAP levels in wild-type intestines with and without a longitudinal opening. As shown below in Figure 3 for reviewers, we found that longitudinal incision decreases nuclear YAP levels, consistent with a role for centripetal forces in activating YAP in the developing intestine. *Edn1*, *Ano5* and *Ccl11* are all YAP-regulated proliferative genes and their reduced expression is further evidence of decreased YAP activity in *Cilk1*^{-/-} intestine (Figure 4E). As these genes participate in cell proliferation, their reduced expression may reflect how decreased nuclear YAP attenuates proliferation in the intestine. We include comment on the significance of these decreases on page 8 of the revised work.

Figure 3 for reviewers. Nuclear YAP intensities of peri-epithelial mesenchymal (peri-epi-me) cells in wild-type intestinal segments incised longitudinally (WT + cut, n=3) or not (WT, n=3) cultured in Matrigel. For each intestine, the nuclear YAP intensity per cell was averaged from three sections. Values are presented as means \pm SEM. * $p < 0.05$ by paired t test.

Figure S11. Quantification would be helpful. It is difficult to see/assess nuclear YAP in these magnified panels.

As suggested, we have added quantifications of nuclear YAP to the revised Figure S11.

Figure 5. It is interesting that mesenchymal loss of Yap/Taz results in decreased epithelial EdU (similar to mesenchymal loss of *Cilk1*). Perhaps this suggests that a loss of mesenchymal proliferation feeds into decreased proliferation of the epithelium as well— a mechanism to ensure coordinated growth rates of the two layers?

We agree and have added comment on this interesting possibility to the new fourth paragraph of the Discussion section.

P.9 Typo: We propose that, at [as] in other tissues. The different effects...

Thank you for noting this typo. We have corrected the text in the revised manuscript.

Figure 6. What does the muscle look like in the *Cilk* mutants after YAP inhibition. Previous studies showed that YAP needed to be downregulated for muscle development. Is muscle development rescued with YAP inhibition?

We appreciate the suggestion for this interesting experiment. As suggested, we inhibited YAP (using XMU-MP-1) to test whether YAP inhibition might rescue smooth muscle development in E13.5 *Cilk1* mutant intestines. cultured intestine where the circumferential smooth muscle cell layer has formed. As depicted below in Figure 4 for reviewers, inhibiting YAP for two days did not restore smooth muscle patterning to *Cilk1* mutants, as measured by α SMA immunofluorescence. We surmise that, as noted by the reviewer, downregulation of YAP is important for smooth muscle cell differentiation, but that CILK1 is required for organization of those differentiating smooth muscle cells into a narrowly circumscribed domain of the intestinal wall.

Figure 4 for reviewers. Immunofluorescence staining of E13.5 intestine of indicated genotypes cultured for two days in Matrigel with DMSO or XMU-MP-1 for smooth muscle (α SMA, green) and nuclei (Hoechst, blue). Scale bars, 25 μ m.

Dr. Nicolas R. Chevalier's review:

In their study, Ying Yang and co-authors examine the biomolecular and mechanobiological cues giving rise to intestinal and oesophageal elongation in the fetal mouse embryo (E13.5-E18.5). Using an impressive array of mutants, they find that cilia located on mesenchymal cells of the intestine are required for gut elongation, with the mutant being down by 40-60% in length compared to controls. Cilia activity is found to be necessary along with hedgehog (HH) to pattern a well-defined, circumferentially aligned smooth muscle ring. Malformation of this smooth muscle ring via a cilia, a HH-pathway defect, or by localized expression of diphtheria toxin results in growth defects. The measurement of opening angles of gut segments shows that the SMA ring contributes to residual circumferential stress, and that this contribution is reduced in all genetic manipulations that resulted in a malformed SMA ring. Yap levels are found to be decreased in the mutants (*Cilk1*^{-/-}, *Dermo1CreSmo* lox/lox), and conditional YAP KO in the mesenchyme results in shorter, less proliferating guts than controls. Boosting YAP signaling with XMU-MP-1 in *Cilk1*^{-/-} intestines rescued elongation in culture. The authors conclude on the following sequence of events: the epithelium produces HH, which is transduced in the mesenchyme by the cilia to give rise to the circumferential smooth muscle ring, which produces force, activating YAP, and activating proliferation and gut lengthening. The authors also demonstrate that embryonic mouse intestines can be grown ex-vivo in matrigel (almost 2-fold elongation after 2-day culture).

I congratulate the authors on this very clear, rigorous, technically sound, exhaustive biomolecular investigation of gut growth in the fetal mouse. The results reinforce several recent studies showing that smooth muscle generated forces are key to gut and lung morphogenesis. I have the following comments:

- my main concern pertains to the fact that the authors do not explain why the force-YAP-proliferation pathway outlined should lead to an elongation of the organ, rather than to its thickening (diameter increase). Khalipina et al. ("Smooth muscle contractility causes the gut to grow anisotropically" *J. Royal Soc. Interface* 2019) have outlined how the Poisson effect (due to the incompressibility of the gut interior) transforms centripetally directed

circumferential smooth muscle pressure (tonic or phasic) into elongation of the embryonic gut. Alternatively, the passive (non-muscle) anisotropic biomechanical characteristics of the gut can favor cell proliferation along the longitudinal axis; these anisotropic properties have also been characterized in Khalipina et al. for the embryonic chicken gut. Both of these mechanisms should be discussed because they complement the author's results.

Thank you for these interesting thoughts, Dr. Chevalier. We concur that investigating how mechanical forces and YAP activation promote longitudinal growth rather than radial expansion needs further study. Possibilities include orientation of cell division and directed cell migration along the longitudinal axis. As described above, we now include additional data to the revised manuscript indicating that CILK1 is especially critical for proliferation along the longitudinal axis. However, these data do not indicate that the circumferential smooth muscle does not promote elongation through the transformation of centripetal forces into elongation or, indeed, other potential mechanisms such as force-directed migration of cells along the longitudinal axis to lead to preferential growth along the longitudinal axis. We have added these points to a new paragraph included in the Discussion of the revised manuscript.

- the embryonic mouse gut undergoes extensive elongation before the smooth muscle differentiates: at E12.5, before any signs of α -SMA can be uncovered, it is already an anisotropic 1 cm long – ~ 200 μ m diameter tube. It thus appears that the force-YAP-proliferation pathway is only the later – fetal - part of the story behind intestinal elongation. The authors should point this out and discuss what mechanisms might underlie early embryonic elongation, prior to SMA differentiation. Are other forces present at early stages that trigger YAP nuclear translocation, e.g., vitelline duct tension? Or could the YAP switch be “off” in early gut morphogenesis?

Alas, the mouse mutants we examined do not affect intestinal elongation before E13.5 during this phase of early development before the smooth muscle differentiates. Indeed, it is likely that, as Dr. Chevalier proposes, either other forces, such as vitelline duct tension, contribute during these stages, or that intestinal elongation is independent of mechanical forces and/or YAP activity at this early phase. Unfortunately, the genetic models we have generated do not shed light on these fascinating possibilities. As suggested, we have added comment about smooth muscle-independent mechanisms of early intestinal elongation to the revised Discussion.

- the authors should discuss whether cilia- of HH-dependent mutations have been documented in short bowel syndrome.

To the revised manuscript, we have added discussion of a recent study reporting that human mutations in *DYNC2LI1*, encoding a component of the ciliary retrograde motor essential for HH signaling, causes congenital short gut¹².

- p.3 the authors should add representative pictures at a given age of B9d1, Inpp5e and Tctn3 mutant guts (whole mount) in Fig.S2 (perhaps in the blank space next to S2B), from which the morphological data of S2D-F was retrieved.

As there is not sufficient space to include these images Figure S2, we include these data for you. If you deem them important, perhaps we can include them as an additional supplementary figure.

Figure 5 for reviewers. Light micrographs of E16.5 *B9d1*, E15.5 *Inpp5e* and E15.5 *Tctn3* intestines of the indicated genotypes and from which Figure S2D-F is derived. Scale bar, 1mm. Scale bars, 1mm.

- it is interesting that guts could be grown in matrigel, while one would think that the elastic gel would act as a mechanical barrier to elongation. Have the authors tried growth in plain liquid BGJb medium, can they comment on the necessity for a gel? The authors should also add the reference to the first paper that reported embryonic mouse gut growth using this medium, Duh. et al. “EGF regulates early embryonic mouse gut development in chemically defined organ culture”, *Pediatr. Res.* 2000.

Thank you for the interesting thoughts. We did test intestinal growth in liquid BGJb medium and found that it did not support robust elongation. Thus, we favor a model in which mechanical forces imposed by a stiff matrix promote intestinal elongation. As suggested, we added reference to Duh et al. in the revised manuscript.

- p.6 “Smooth muscle was positive for phospho-MLC at E18.5, but negative at E13.5 (Figure S9), suggesting that the circumferential smooth muscle promotes intestinal elongation before it becomes contractile.” is not correct, because propagating contractile waves can be observed propagating along the whole midgut at E13.5 (but not at E12.5) – the authors can verify this by time-lapse imaging E13.5 guts or by checking this reference: R. Roberts et al. “The first intestinal motility patterns in fetal mice are not mediated by neurons or interstitial cells of Cajal”, *J. Phys.* 2010. The circular smooth muscle has tonic (time-independent) and phasic (dynamic) contractile components that are both active in generating force on the gut interior (see Khalipina et al.), and thus in activating YAP – more on that below.

Thank you for this correction. Indeed, we have also observed that both the frequency and amplitude of the intestinal smooth muscle contraction increases as development proceeds. We have modified the text in page 7 in the revised manuscript to note that circular smooth muscle contraction can contribute to force generation in the developing intestine and its elongation.

- The opening angles (Fig.3, p.7) are measured at room temperature in DPBS without calcium, a situation where the smooth muscle is not able to contract: it is thus passive (non-

muscle) stress due to the shear presence of this layer of tensed aligned cells & ECM that is measured. The authors should clarify that this is however only a minor component of the stress, the other – main one – being the active tonic and phasic force generated when SMA is active (as in a developing mouse embryo at 37°C). Although these have not been measured by the authors, the result would very probably be that the various mutants generate less active stress. This should be pointed out and the authors should replace the term “residual stress” in the remainder of the manuscript by “force” (example p.7: “raised the interesting possibility that force produced by the circumferential smooth muscle promotes longitudinal growth”).

As suggested, we clarified that opening angles measured in DPBS without calcium remove the effects of muscle contraction in page 7 of the revised text. As also suggested, we have replaced the term "residual stress" in the remainder of the revised manuscript when discussing circumstances in which muscle contraction may contribute to forces within the developing intestine.

- the unaltered smooth muscle layer of the E13.5 *Yap*^{lox/lox} could usefully be moved from S12B to Fig.5 (a neat blank space is left).

As further suggested, we moved the images of E13.5 *Yap*^{lox/lox} *Taz*^{lox/+} and *Dermo1*^{Cre} *Yap*^{lox/lox} *Taz*^{lox/+} intestines stained for YAP and smooth muscle to Figure 5. In addition, in the revised manuscript, we quantified αSMA staining intensity distribution across the radial thickness of the gut tube. These results bolster the finding that *Dermo1*^{Cre}-mediated depletion of YAP does not alter smooth muscle development. These new data are included as Figure S12B.

- p.9 please clarify what is “GLI”

GLI refers to the GLI transcription factors that are the effectors of HH signaling. Perhaps unfortunately, the GLI designation historically was given as they were thought to participate in glioma biology. As it remains uncertain whether GLI proteins are relevant to glioma biology, we clarified the GLI term as "GLI transcription factors, effectors of HH signaling" in the revised manuscript.

- the title of the manuscript is inaccurate: experiments have been performed on the small intestine and the esophagus, i.e., on the gastrointestinal tract. Other smooth muscle lined tubular organs are of course the blood vessels, and although an extension of the mechanism outlined sounds exciting, it remains speculative. Perhaps a title more reflective of the biochemical effort that went in the study on the HH-cilia-smooth muscle-force-YAP pathway would be more informative.

We concur with the critique. Thus, we have changed the title to "Ciliary Hedgehog signaling patterns the digestive system to generate mechanical forces driving elongation."

Thank you for your extensive help with this manuscript. Again, I deeply appreciate your efforts and suggestions, which have substantially improved our work, as well as the opportunity to submit a revised manuscript to you. Please let me know if I can provide you with any additional information or answer any further questions you might have.

References

1. Lambricht, L. *et al.* The type and composition of alginate and hyaluronic-based hydrogels influence the viability of stem cells of the apical papilla. *Dent Mater* **30**, e349–61 (2014).
2. Anguiano, M. *et al.* Characterization of three-dimensional cancer cell migration in mixed collagen-Matrigel scaffolds using microfluidics and image analysis. *PLOS ONE* **12**, e0171417 (2017).
3. Motte, S. & Kaufman, L. J. Strain stiffening in collagen I networks. *Biopolymers* **99**, 35–46 (2013).
4. Dewitt, D. D., Kaszuba, S. N., Thompson, D. M. & Stegemann, J. P. Collagen I-matrigel scaffolds for enhanced Schwann cell survival and control of three-dimensional cell morphology. *Tissue Engineering Part A* **15**, 2785–2793 (2009).
5. Soofi, S. S., Last, J. A., Liliensiek, S. J., Nealey, P. F. & Murphy, C. J. The elastic modulus of Matrigel as determined by atomic force microscopy. *J Struct Biol* **167**, 216–219 (2009).
6. Huycke, T. R. *et al.* Genetic and Mechanical Regulation of Intestinal Smooth Muscle Development. *Cell* **179**, 90–105.e21 (2019).
7. Mao, J., Kim, B.-M., Rajurkar, M., Shivdasani, R. A. & McMahon, A. P. Hedgehog signaling controls mesenchymal growth in the developing mammalian digestive tract. *Development* **137**, 1721–1729 (2010).
8. Wang, C., Li, J., Meng, Q. & Wang, B. Three Tctn proteins are functionally conserved in the regulation of neural tube patterning and Gli3 processing but not ciliogenesis and Hedgehog signaling in the mouse. *Dev. Biol.* **430**, 156–165 (2017).
9. Liu, A., Wang, B. & Niswander, L. A. Mouse intraflagellar transport proteins regulate both the activator and repressor functions of Gli transcription factors. *Development* **132**, 3103–3111 (2005).
10. Wijgerde, M., McMahon, J. A., Rule, M. & McMahon, A. P. A direct requirement for Hedgehog signaling for normal specification of all ventral progenitor domains in the presumptive mammalian spinal cord. *Genes & Development* **16**, 2849–2864 (2002).
11. Yee, L. E. *et al.* Conserved Genetic Interactions between Ciliopathy Complexes Cooperatively Support Ciliogenesis and Ciliary Signaling. *PLoS Genet.* **11**, e1005627 (2015).
12. Bryson, L. J., Flynn, D. M., Sabharwal, A., Ahmed, S. F. & Kinning, E. A child with congenital short gut associated with DYNC2LI1 ciliopathy. *Clinical Dysmorphology* **30**, 66–68 (2020).

Reviewers' Comments:

Reviewer #1:

Remarks to the Author:

The authors have carefully considered my suggestions and added new data, text changes, and figure changes that significantly improve the paper. I am happy to support publication of this excellent work.

Reviewer #2:

Remarks to the Author:

The authors have sufficiently addressed previous concerns noted in the first peer review and this recommender would be pleased to see this manuscript published in Nature Communications. The only new suggestion is to please review the following sentence on page 5 to clarify the meaning of this sentence: "These results demonstrate that mesenchymal CILK1 is essential for proliferation of both the mesenchyme and its associated epithelium, suggesting that the intestinal mesenchyme has important roles in promoting both proliferation and elongation." Did you mean to say that it has essential roles in promoting proliferation and elongation of both the epithelium and the mesenchyme?

Reviewer #3:

Remarks to the Author:

The authors have done a good job in the revision. The possibility of distinguishing longitudinal versus radially dividing cells by tubulin (Fig.S1 E-F) staining is very interesting. I have the following comments:

- Major points concerns the added experiment Fig.S10B that Y-27632 ROCK inhibitor does not alter intestinal lengthening in culture, from which the authors conclude that smooth muscle contraction is not essential for elongation. The authors need to test whether this molecule is efficient at halting intestinal contractions by time lapse imaging spontaneous contractions in E13.5 or E14.5 guts in physiological conditions. The K_i of Y-27632 for MLCK (indispensable for smooth muscle contraction) is 250 μM , way above the concentration used in the study (10 μM). Nifedine or nicardipine 10 μM would be an alternative, certified way of halting the contractions.

The effect of Y-27632 on opening angles could also be evaluated. In our experience in embryonic mice, smooth muscle contraction inhibition (with nicardipine) reduces the opening angle approximately by the same amount than the various mutants presented in Fig.3D-F. Which would, following the logic of the article, decrease YAP and lengthening of the gut ...

Minor:

- there is a typo in Fig.S1-F ("radical").
- P.7 "in the absence of supplemented extracellular calcium at room temperature"
- P.11 please replace "the external physical force exerted by the matrix may compensate for the reduced force imposed by Cilk1 mutant intestines" by "the stiffness of the matrix may compensate for the reduced force generated by Cilk1 mutant intestines" "because the matrix as such does not exert a force on the intestine – it is only if cells start migrating or proliferating in the gel (as during the elongation) that they can feel this stiffness and generate a force in response to this stiffness. That being said, this result is surprising !

Best regards,

Dr. Nicolas R. Chevalier

Dear Reviewers,

I and my co-authors again thank you for your thoughtful comments. We were heartened to read that reviewers 1 and 2 support publication and that Dr. Chevalier commented the completion of a good job for the revision. We include descriptions below responding to the remaining points. Critiques are delineated in blue font.

REVIEWER COMMENTS

Reviewer #1:

The authors have carefully considered my suggestions and added new data, text changes, and figure changes that significantly improve the paper. I am happy to support publication of this excellent work.

Reviewer #2:

The authors have sufficiently addressed previous concerns noted in the first peer review and this recommender would be pleased to see this manuscript published in Nature Communications.

The only new suggestion is to please review the following sentence on page 5 to clarify the meaning of this sentence: "These results demonstrate that mesenchymal CILK1 is essential for proliferation of both the mesenchyme and its associated epithelium, suggesting that the intestinal mesenchyme has important roles in promoting both proliferation and elongation." Did you mean to say that it has essential roles in promoting proliferation and elongation of both the epithelium and the mesenchyme?

Yes, we did mean to say that mesenchymal CILK1 is important for promoting proliferation and elongation in both the epithelium and mesenchyme. We have revised the sentence on page 5 to make that point clearer.

Reviewer #3:

The authors have done a good job in the revision. The possibility of distinguishing longitudinal versus radially dividing cells by tubulin (Fig.S1 E-F) staining is very interesting. I have the following comments:

- Major points concerns the added experiment Fig.S10B that Y-27632 ROCK inhibitor does not alter intestinal lengthening in culture, from which the authors conclude that smooth muscle contraction is not essential for elongation. The authors need to test whether this molecule is efficient at halting intestinal contractions by time lapse imaging spontaneous contractions in E13.5 or E14.5 guts in physiological conditions. The K_i of Y-27632 for MLCK (indispensable for smooth muscle contraction) is 250 μ M, way above the concentration used in the study (10 μ M). Nifedine or nicardipine 10 μ M would be an alternative, certified way of halting the contractions.

We discussed this experiment with the editor. As the main focus of the paper is on the role of ciliary Hedgehog signaling in patterning the gut wall, the role of smooth muscle contractions in elongation is interesting but not the central subject. Therefore, we have addressed your point with changes to the text. More specifically, we clarify that these data in no way rule out roles for smooth muscle contraction as a driver of intestinal elongation, and while 10 μ M Y-27632 can inhibit muscle contraction as reported previously by Narumiya et al (2000) and Sims et al (2008) among others, we clarify that smooth muscle contraction can contribute to intestinal forces and elongation at stages other than the one we examined and in specific species. Additionally, we explicitly discuss in the revised manuscript's Discussion section data from your group, reported in Khalipina et al (2019), the role of smooth muscle contraction in intestinal elongation.

The effect of Y-27632 on opening angles could also be evaluated. In our experience in embryonic mice, smooth muscle contraction inhibition (with nicardipine) reduces the opening angle approximately by the same amount than the various mutants presented in Fig.3D-F. Which would, following the logic of the article, decrease YAP and lengthening of the gut...

We fully agree that the effect of Y-27632 on opening angles could also be evaluated. Ying may seek to do so for future works focused on the function of smooth muscle contraction on YAP activity and gut lengthening.

Minor:

- there is a typo in Fig.S1-F (“radical”).

Thank you for pointing out this typo. We have fixed it.

- P.7 “in the absence of supplemented extracellular calcium at room temperature”

We have made the recommended change to the text.

- P.11 please replace “the external physical force exerted by the matrix may compensate for the reduced force imposed by Cilk1 mutant intestines” by “the stiffness of the matrix may compensate for the reduced force generated by Cilk1 mutant intestines” “because the matrix as such does not exert a force on the intestine – it is only if cells start migrating or proliferating in the gel (as during the elongation) that they can feel this stiffness and generate a force in response to this stiffness.

Thank you for the correction. We have also made this change to the text.

Thank you for your thoughtful comments and valuable insights, Dr. Chevalier. We are deeply grateful for your corrections. Please let me know if I can provide you with any additional information or answer any further questions you might have.

Reviewers' Comments:

Reviewer #3:

Remarks to the Author:

Dear authors,

Thank you for the revised manuscript. My feeling on the Y-27632 experiment is that it is not quite consistent with the theory developed by the authors. If it does inhibit the smooth muscle, then it should relax not only the contractions, but also the tone (what is called in the manuscript the "residual stress due to smooth muscle"), and have a similar effect on reducing gut lengthening as the different mutants presented in the article. An alternative is that Y-27632 does not affect the tone / contractions, in spite of the effects reported in other systems (pig coronary artery in Narumiya et al., human esophageal muscle in Sims et al.); my experience is that inhibitors reported to work for some smooth muscles do not necessarily do the job for the embryonic gut (an example: wortmannin). Without wanting to further delay publication of this excellent work I suggest that the authors modify l.321: "Our data suggest that smooth muscle contraction may not be essential for mouse intestinal elongation at E13.5. Y-27632 has been shown to inhibit muscle contraction at this concentration in multiple systems 59,60, but its inhibitory effect on contractions and smooth muscle tone in the embryonic mouse gut would require confirmation."

Best regards,

Nicolas Chevalier

Dear Dr. Chevalier,

Thank you for your additional helpful suggestions. It was rewarding to read that you deemed our study “excellent.” We have incorporated your suggestion and modified the sentence in the Results section to read:

As Y-27632 has been shown to inhibit muscle contraction at this concentration in multiple systems (but not confirmed here in the embryonic mouse gut)^{1,2}, our data suggest that smooth muscle contraction may not be essential for mouse intestinal elongation at E13.5.

Thank you for your insightful comments – they have been a great guide to strengthening this work and we are in debt to you!

1. Narumiya, S., Ishizaki, T. & Uehata, M. Use and properties of ROCK-specific inhibitor Y-27632. *Methods Enzymol* **325**, 273–284 (2000).
2. Sims, S. M., Chrones, T. & Preiksaitis, H. G. Calcium sensitization in human esophageal muscle: role for RhoA kinase in maintenance of lower esophageal sphincter tone. *J Pharmacol Exp Ther* **327**, 178–186 (2008).